



# Analysis of a newly homogenised ozonesonde dataset from Lauder, New Zealand

Guang Zeng[1], Richard Querel[2], Hisako Shiona[3], Deniz Poyraz[4], Roeland Van Malderen[4], Alex Geddes[2], Penny Smale[2], Dan Smale[2], John Robinson[2], and Olaf Morgenstern[1]

[1]National Institute of Water and Atmospheric Research (NIWA), Wellington, New Zealand
[2]National Institute of Water and Atmospheric Research (NIWA), Lauder, New Zealand
[3]National Institute of Water and Atmospheric Research (NIWA), Christchurch, New Zealand
[4]Royal Meteorological Institute, Uccle, Belgium

**Correspondence:** Guang Zeng (guang.zeng@niwa.co.nz)

**Abstract.** This study presents an updated and homogenised ozone time series covering 34 years (1987-2020) of ozonesonde measurements at Lauder, New Zealand, and derived vertically resolved ozone trends. Over the period of 1987-1999, the ozone trends in the homogenised ozone data are predominantly negative from the surface to $\sim$30 km, ranging from $\sim -2$ to $-12\%$ decade$^{-1}$, maximising at around 12-13 km. These negative trends are statistically significant at 95% confidence level below 5 km and above 17 km. For the post-2000 period, ozone at Lauder shows negative trends in the stratosphere (but the trends are only statistically significant above 17 km), maximising just below 20 km ($\sim -5\%$ decade$^{-1}$), despite stratospheric chlorine and bromine from ozone-depleting substances (ODSs) both declining in this period. In the troposphere, the ozone trends change from negative for 1987-1999 to positive in the post-2000 period. The post-2000 ozone trends from the ozonesonde measurements compare well with those from a low-vertical resolution Fourier-transform infrared spectroscopy (FTIR) ozone time series. A multiple-linear regression analysis indicates that anthropogenic forcing plays a significant role in driving the significant negative trend in the stratospheric ozone at Lauder, in which the effect of greenhouse gas (GHG)-driven dynamical and chemical changes is reflected in the significant positive trends in tropopause height and tropospheric temperature, and significant negative trends of stratospheric temperature observed at Lauder. The interannual variation in lower stratospheric ozone is largely explained by the variation in tropopause height at Lauder, which is highly anti-correlated with stratospheric temperature and correlated with tropospheric temperature. Furthermore, the impact of ODSs and GHGs on ozone over Lauder is assessed in a chemistry-climate model using a series of single forcing simulations. The model simulations show that the predominantly negative modelled trend in ozone for the 1987-1999 period is driven not only by ODSs, but also by increases in GHGs with large but opposing impacts from methane (positive) and $CO_2$ (negative), respectively. Over the 2000-2020 period, although the model underestimates the observed negative ozone trend in the lower stratosphere but clearly shows that $CO_2$-driven dynamical changes have had an increasingly important role in driving ozone trends in this region.



## 1 Introduction

Ozone ($O_3$) plays a central role in atmospheric chemistry and the radiation budget. The stratospheric ozone layer protects life on Earth by preventing harmful ultra-violet radiation from reaching the surface. Stratospheric ozone is also a natural
source of tropospheric ozone via cross-tropopause transport; it accounts for around 30% of tropospheric ozone production (Lelieveld and Dentener, 2000). Since the late 1970s, due to the release of man-made ozone depleting substances (ODSs), Southern-Hemisphere stratospheric $O_3$ changes are dominated by Antarctic ozone depletion leading to negative trends in stratospheric ozone (e.g., WMO, 2014, 2018). Due to the successful implementation of the Montreal Protocol (MP) in 1987 and its subsequent amendments, concentrations of ODSs have been declining. The most recent assessment (WMO, 2022)
confirms that upper-stratospheric ozone is recovering, in agreement with model simulations (Godin-Beekmann et al., 2022; Zeng et al., 2022).

However, while the ODSs are declining, the future evolution of ozone depends critically on changes in greenhouse gases (GHGs). For example, decreases in stratospheric temperature caused by increasing $CO_2$ and other GHGs will accelerate stratospheric ozone recovery (Randeniya et al., 2002; Rosenfield et al., 2002). In the tropical lower stratosphere, climate change in-
35 creases tropical upwelling, leading to less time for $O_3$ production and hence decreasing $O_3$ in this region (Eyring et al., 2010). As a result, both observations and models indicate a small but uncertain decrease of ozone in the tropical lower stratosphere which is consistent with the Brewer-Dobson circulation (BDC) change driven by increases in greenhouse gases (WMO, 2022). In both mid-latitude regions, the combined satellite stratospheric ozone trends are generally negative albeit non-significant over the period 2000-2020, but such observed trends are not reproduced by either CCMI-1 or AerChemMIP model simulations
which show generally non-significant positive trends in these regions (Godin-Beekmann et al., 2022; Zeng et al., 2022; WMO, 2022). The ozone distribution is typically affected by large dynamical variability in the lowermost stratosphere, limiting any attribution to anthropogenic factors. Furthermore, future changes of stratospheric $O_3$ could also significantly impact tropospheric $O_3$ and potentially air quality through stratosphere-troposphere exchange (STE), especially in the Southern Hemisphere where the stratospheric ozone influx plays a larger role in the tropospheric ozone budget, relative to in-situ ozone formation, than in
the more polluted Northern Hemisphere (e.g., Zeng et al., 2010; Hegglin and Shepherd, 2009).

High vertical resolution ozone measurements are key to understanding the impact of various anthropogenic forcings on ozone changes, especially in the upper troposphere and the lower stratosphere where the large dynamical variability may obscure any attempts of attribution using the models. The high vertical resolution ozonesonde measurements are well-positioned to detect changes in ozone from the surface to around 35 km. An extensive ozonesonde measurement network exists throughout the
50 Northern Hemisphere, but, it is sparse in the Southern Hemisphere (SH). Lauder, New Zealand (45°S, 170°E, 370 m above sea level), a clean rural site that is representative of the SH mid-latitude background atmosphere, is a primary member of the Network for the Detection of Atmospheric Composition Change (NDACC). The Lauder ozonesonde measurements started in 1986 and continue to provide weekly high-resolution vertically resolved ozone data from the surface to around 35 km; this is of particular relevance to detecting long-term changes in both the stratospheric and tropospheric ozone in the SH clean
background air (Oltmans et al., 2006, 2013; Zeng et al., 2017).



Recently, the Lauder ozonesonde data have been subjected to a homogenisation process under the guidance of the Ozonesonde Data Quality Assessment (O3S-DQA) activity (Smit and the O3S-DQA panel, 2012), which is part of the SPARC/IO3C/IGACO-O3/NDACC) (SI2N) initiative (Harris et al., 2011, 2012). Homogenisation is designed to produce consistent datasets with reduced uncertainties and offsets in long-term ozone vertical profiles that arise from instrumental and operating procedure

changes over the observational periods. Any heterogeneities the data have can adversely affect trend calculations. Many other ozonesonde measurement sites have gone through this homogenisation process (Tarasick et al., 2016; Van Malderen et al., 2016; Thompson et al., 2017; Sterling et al., 2018; Witte et al., 2017, 2018, 2019; Ancellet et al., 2022), and we have applied the same procedure to homogenise the Lauder ozonesonde timeseries between August 1986 (when the observation started) and June 2021, although we only take the data from January 1987 to December 2020 for analysis here. The post-2000 homogenised

Lauder ozone dataset was included by Godin-Beekmann et al. (2022) in their evaluation of near-global ($60^{o}$S–$60^{o}$N) stratospheric ozone profile trends from satellite and multiple ground-based instruments, along with datasets from several other ozone measurements, using an updated version of the Long-term Ozone Trends and Uncertainties in the Stratosphere (LOTUS) regression model (LOTUS, 2019). Godin-Beekmann et al. (2022) show that the negative ozone trends in the lower stratosphere from the Lauder ozonesonde timeseries were exceedingly large in absolute terms in comparison with the trends calculated from

the satellite data and from other instruments at the same site.

In this paper, we present the homogenised Lauder ozonesonde record covering the whole observational period of 1987-2020, and evaluate vertically resolved ozone trends from the surface to 30 km for both the pre-1999 and the post-2000 periods, and contrast these with the data series without homogenisation. The post-2000 ozone trends are also compared to trends obtained from the co-located Fourier-transform infrared spectroscopy (FTIR) measurements, which has been updated from the dataset

used in Godin-Beekmann et al. (2022). We aim to identify the dominating forcing that drives ozone trends at Lauder using a multiple linear regression (MLR) model analysis. We will assess the role of ODSs and GHGs (including methane, $N_2O$, and $CO_2$) in driving ozone changes over the last few decades and into the near future using simulations from a chemistry-climate model, in relation to changes in ozone trends at Lauder, representative of the background $O_3$ changes in the Southern Hemisphere mid-latitudes. In the next section, we describe the homogenised ozone time series, construct the MLR model, and

describe the CCM simulations. We then present the results and discussions in Sect. 3. Conclusions are drawn in section 4.

## 2 Data and regression model

### 2.1 Homogenised ozonesonde records

Weekly electrochemical cell (ECC) ozonesondes have been launched in tandem with radiosondes at Lauder since August 1986, measuring profiles of ozone, temperature, pressure, humidity, and wind speeds and directions from the surface up to about 35

85 km (Boyd et al., 1998; Bodeker et al., 1998). The ECC used for ozone sounding at Lauder are the Science Pump Corporation (SPC) series 4A/5A/6A (before 1996) and the Environmental Science (EnSci) Z series (after 1996), although there are some overlap period when both types were used. These ECC series were operated with a 1.0% buffered potassium iodide (KI)





cathode solution until July 1996 and a 0.5% KI solution from August 1996 until present. These changes are relevant to the homogenisation process and are detailed in Table 1.

The homogenisation procedure, described in the Assessment of Standard Operating Procedures for Ozonesondes (ASOPOS 2.0) documentation (Smit et al., 2021) and in the Ozonesonde Data Quality Assessment (O3S-DQA) activity (Smit and the O3S-DQA panel, 2012), was applied to the Lauder ozonesonde timeseries, available at NDACC. These NDACC data, named "uncorrected data" hereafter, have been obtained by converting the raw currents measured with an ozonesonde to ozone partial pressures by subtracting a measured background current, using a conversion efficiency of 1.0, the measured pump temperature

and pump flow rate measured prior to launch in the lab, and correcting for the pump efficiency decrease with increasing altitudes. The O3S-DQA homogenization, however, add corrections to the pump temperature, the pump flow rate (due to the moistening effect), and the background current (avoiding too high values) on top, and uses a set of transfer functions applied to the conversion efficiency to remove biases due to changes in the instrument or operating procedures. For example, a transfer function is needed for the change of sensing solution because there was a 2-year period when the EnSci ECCs started to be used, but

with the 1% solution, rather than the 0.5% solution which has become the recommendation for the EnSci ECCs. The re-process of the Lauder data according to the O3S-DQA guidelines were carried out by the HEGIFTOM working group (Harmonization and Evaluation of Ground Based Instruments for Free Tropospheric Ozone Measurements, https://hegiftom.meteo.be) within the TOAR-II (Tropospheric Ozone Assessment Report phase II, https://igacproject.org/activities/TOAR/TOAR-II) initiative. Further details regarding the corrections are summarised in Appendix A.

In this study, we include a total of 1958 ozonesonde flights between August 1986 and June 2021, which the data have been homogenised. Both homogenised and the uncorrected datasets have been post-processed for trend calculations and the regression analysis (in the case of homogenised data). Linear piecewise regression was applied to interpolate the recorded ozonesonde data vertically to a 1 km resolution grid. We then exclude some points with extreme ozone values (i.e. any ozone values that are outside the 3 standard deviation range for the whole time series at each vertical level) to create monthly means

by averaging the data available for that month at each re-gridded vertical level. We calculate the ozone trends in two periods, i.e., the 1987-1999 and the 2000-2020 periods for grouped vertical layers from the surface to 30 km.

## 2.2 Regression model

We construct a multiple linear-regression (MLR) model to identify the dominant factors that are associated with $O_3$ variations and trends. The homogenised $O_3$ mixing ratios are averaged over eight layers (0–1.5, 1.5–3, 3–6, 6–9, 9–12, 12–15, 15–20, and

20–25 km). We then construct regression models for each layer. In total, the regression models include nine terms representing the Solar Index (SI) which captures solar variability and is defined by the solar radio flux at 10.7 cm, the Multivariate El Niño Southern Oscillation Index (MEI), the Quasi-Biennial Oscillation at 30 hPa and 10 hPa, respectively ($QBO_{30}$ and $QBO_{10}$), tropopause height ($HT_{Trop}$), surface relative humidity ($RH_{surf}$), aerosol optical depth (AOD), and the equivalent effective stratospheric chlorine (EESC). The EESC is defined as a relative measure of the potential for stratospheric ozone depletion

that combines the contributions of chlorine and bromine from surface observations from ODSs (Newman et al., 2007), and is calculated based on the ozone-depleting substances from the Coupled Model Project 6 (CMIP6) historical (until 2014) and





the Shared Socio-economic Pathway (SSP245) (for 2015-2021) scenarios (Meinshausen et al., 2017). Surface humidity are measured by the radiosonde that has a humidity sensor. We define the tropopause based on the WMO lapse rate definition WMO (1957), which is calculated using the co-measured temperature data of each ozonesonde flight. The regressed $O_3$ anomaly is

expressed as

$$Ozone(t) = a_1 \cdot t + a_2 \cdot SI(t) + a_3 \cdot SOI(t) + a_4 \cdot QBO_{10}(t) + a_5 \cdot QBO_{30}(t)$$
$$+ a_6 \cdot EESC(t) + a_7 \cdot AOD(t) + a_8 \cdot HT_{Trop}(t) + a_9 \cdot RH_{surf}(t) + \epsilon(t). \tag{1}$$

Here $Ozone(t)$ is the monthly mean ozone timeseries minus its mean annual cycle, $\epsilon$ is the regression residual, minimized in the RMS, $a_1$ is the linear trend (or L-trend) and $a_{2-9}$ are the regression coefficients for the corresponding regressors, all normalized to vanishing means and unit standard deviation. All forcings used in regression are summarised in Table 2, and their

time series are displayed in Fig 6. The tropopause height anomaly is de-trended because it is well-known to be influenced by increasing GHGs (whose influence is already encapsulated in the linear trend term $a_1 t$). The surface humidity is not de-trended as it does not have a significant linear trend over the observation period. All other regressors represent external forcings which are not coupled to the GHGs and none of them have significant trends therefore they are not de-trended. The two QBO indices are orthogonalized w.r.t. each other. Observations as well as basis functions are smoothed using a 12-months boxcar filter.

## 2.3   Chemistry-climate model simulations

We use the NIWA-UKCA model simulations from the Chemistry-Climate Model Initiative project (CCMI-1: Eyring et al. (2013), Morgenstern et al. (2017)) to assess the impact of the major anthropogenic forcings, including greenhouse gases (GHGs) and ozone depleting substances (ODSs), on ozone changes at Lauder. The Lauder ozonesonde measurements cover both the ozone depletion and the recovery periods of 1987-2020. Global chemistry-climate models (CCMs), including NIWA-

UKCA (Morgenstern et al., 2009; Zeng et al., 2015, 2017), generally have coarse resolution. Therefore it is not ideal to use the simulations from the CCMs for reproducing the observed trends at a specific location. Instead, the CCMs can be used to attribute the trends to various forcings on a wider spatial and temporal scale. Here, we calculate the ozone trends using the NIWA-UKCA simulations to gauge the impacts of GHGs and ODSs on ozone changes at Lauder over the observational period in a limited area covering Lauder (averaged over 160-180°E and 40-50°S). We also show the simulation results on a

global scale in context. The CCMI-1 simulations from NIWA-UKCA used here consist of the all forcing coupled atmosphere-ocean reference experiment "RefC2", covering the simulation period of 1960-2100 (we keep the same experiments naming convention as defined by Eyring et al. (2013)) and its corresponding single forcing sensitivity simulations (sen-C2-fODS, sen-C2-fGHGs, sen-C2-fCH$_4$), and sen-C2-fN$_2$O) in which ODSs, combined GHGs, methane (CH$_4$), and N$_2$O are fixed at their 1960's levels, respectively. The impact of each single forcing on ozone is derived from the differences in ozone between

the reference simulation and the corresponding single forcing simulation. We can directly assess the impacts of changes in ODS, combined GHGs, methane, and N$_2$O on ozone trends, based on available simulations. However, no simulation was performed to directly assess the impact of CO$_2$ within CCMI-1; instead, it will be derived from the available fixed methane, N$_2$O, and combined GHGs experiments to subtract the impacts of methane and N$_2$O from the impact of combined GHGs



(Morgenstern et al., 2018). Unlike the ODSs, which peaked in the late 1990s, GHGs (including $CO_2$, methane, and $N_2O$) are mostly monotonically increasing. The impacts of GHGs changes on future ozone evolution are expected to be dominant while the ODSs are declining. We therefore also separately examine the changes in modelled ozone over the period of 2000-2020. The detailed description of the model and experiments can be found in Morgenstern et al. (2018) and in Zeng et al. (2017) and the references therein.

## 3 Results

### 3.1 Homogenised versus uncorrected ozonesonde time series

The homogenised and the uncorrected datasets are directly compared without any temporal interpolation, but are both interpolated in the vertical to 1 km grid using piecewise linear regression for each profile. Figure 1 shows the percentage difference between vertical ozone profiles from the two datasets for all flights. Overall, corrections lead to mostly increased ozone values in the homogenised time series, reaching 6 to over 10% before 1995 due to the pump temperature correction (Figure A1(3)). The pump flow rate correction results in a uniformly positive effect of less than 2% in general (Figure A1(4)). There are scattered increases in ozone in the homogenised time series compared to the uncorrected data, especially between 2012 and 2015 when a modified background current correction is applied (Figure A1(2)). The effect of changes to the concentration of the KI solution on the conversion efficiency (Figure A1(1)) is mainly negative between 1994 and 1996 but positive in the beginning of the time series (1986), when a smaller cathode sensing solution amount has been used (2.5 ml instead of 3 ml). The correction procedure and the impact of each correction are described in more detail in Appendix A.

The differences between the homogenised and the uncorrected monthly mean ozone time series are calculated excluding outliers where ozone is outside the 3 standard deviation interval of all data points for that level (Figure 2). This step removes less than 1% of data points from the monthly mean ozone calculations. Most of these outliers are around 10 km where the ozone is subjected to large dynamical variations. We carry out trend calculations based on the monthly mean ozone values (section 3.2).

### 3.2 Vertically resolved ozone trends at Lauder

Consequently, there are marked differences in ozone trends in the homogenised data compared to those in the uncorrected data over the 1987-1999 period (Figures 3 and 4). During this period, the vertically resolved trends in the uncorrected data set are slightly negative throughout most of the domain above 10 km and positive below 10 km, although below 25 km these trends are generally insignificant at the 95% confidence level (Figure 3). In contrast, the trends in the homogenised data are negative throughout the domain ($\sim -2$ to $-12\%$ decade$^{-1}$), with most of the trends below 5 km and above $\sim 12$ km being statistically significant at the 95% confidence level. This situation is more consistent with the effect of ozone depletion at mid-latitudes driven by increasing ozone depleting substances (ODSs) over this period. In the post-2000 period, the calculated trends are very similar between homogenised and uncorrected ozone profiles (Figures 3 and 4). Both show significant positive trends of up to



∼2% decade$^{-1}$ in the free troposphere and a significant negative trend of ∼2-6% decade$^{-1}$ above ∼16 km in the stratosphere, which peaks around 18 km. We note that the lower stratospheric ozone negative trend of ∼3-6% decade$^{-1}$ between 15 and 20 km looks markedly smaller in magnitude than the trend presented in Godin-Beekmann et al. (2022) where it exceeds 6-7% decade$^{-1}$ in the same region. Distinct negative trends of ∼2-4% decade$^{-1}$ also exist in the upper troposphere and the lower stratosphere between 8 and 16 km albeit with large statistical uncertainty, highlighting the large dynamical variability

typical for this region. We find that the vertically resolved ozone trends calculated by excluding the outliers from creating the monthly mean ozone values are very similar to the trends calculated by including all data points in the monthly mean. The only difference is that by excluding the outliers we have reduced the trend uncertainties around the 10 km region (not shown).

We compare the homogenised post-2000 ozonesonde data to ozone profiles from the Fourier-transform infrared spectroscopy (FTIR) ozone measurements, from which low-resolution vertical profiles are derived (Vigouroux et al., 2015, e.g.,). The FTIR

data presented here are an updated version of data shown in Godin-Beekmann et al. (2022) based on the retrieval strategy presented in García et al. (2022). Our analysis reveals very good agreement of the ozone trends derived from these two different measurements (Figure 4). The negative trends in the lower stratosphere in both the sonde and the FTIR ozone data are noticeably larger in magnitude than the trends in the satellite data shown in Godin-Beekmann et al. (2022) which have typically insignificant trends of smaller than −2% decade$^{-1}$.

We also examine how seasonal variations in vertical distributions of ozone might have contributed to the overall trends over the observation period. The seasonal ozone anomalies representing austral summer (DJF), autumn (MAM), winter (JJA), and spring (SON) from the homogenised ozonesonde data are shown in Figure (5). It shows that the ozone evolution among all seasons is broadly consistent at the selected vertical levels from the surface to 25 km. The surface ozone in all seasons show decreases until the late 1990s before peaking before ∼2010. The seasonal variation is also weak above 5 km showing

downward trends in all seasons. Some slight differences between seasons are below 5 km, where the DJF trend is weaker before the mid-1990s. Also the JJA ozone seems quite flat around 5 km whereas the DJF ozone has a sharp drop after 2015; it warrants a further investigation with a longer time series into the future.

### 3.3 Drivers of ozone trends at Lauder

#### 3.3.1 Variations and trends explained by regression analysis

In order to identify the drivers of ozone variability and trends, we construct a regression model (eq. 1) to explain the ozone variance. The deseasonalised monthly mean ozone anomalies of the homogenised ozonesonde data are grouped into eight layers from the surface to 25 km. The regression is performed individually for each layer. The independent regressors used in the regression are shown in Figure 6, and the observed and regressed ozone anomalies are shown in Figure 7. The ozone variance explained by the regression is given by the multiple regression coefficient of determination, $R^2$ (table B1). The standardised

individual regression coefficients for each regressor can be used to measure their contributions to the total variance explained at that level (Figure 8 and Table B1). The leading contributions from individual terms to overall regressed ozone variations are demonstrated in Figure 9.





The regression model matches the observed anomalies well, in particular in the stratosphere, the upper troposphere, and near the surface (Figure 7), with $R^2$ ranging from 0.27 to 0.49 in the troposphere and 0.50 to 0.73 in the stratosphere, im-
220 plying that the stratospheric ozone variations and trends are better explained by the MLR model than tropospheric features. Indeed, interannual variations in ozone anomalies in the upper troposphere and the lower stratosphere (9-15 km) are especially well explained by variations in tropopause height. The downward trend in the stratospheric ozone is clearly explained by the significant negative linear trend that represents all quasi-linear, monotonic drivers of change (Figures 8 and 9). Note that the tropopause height anomalies have been de-trended for use in regression here, therefore do not explain the negative trends in
ozone. No significant trends exist in other regressors either.

It is well established that $CO_2$ increases influence temperature, humidity, and circulation, which in turn affect ozone chemistry and transport (Brasseur and Hitchman, 1988; Butchart et al., 2006; Fleming et al., 2011). Warming in the troposphere and the cooling in the stratosphere due to the increase in $CO_2$ drives the increase in tropopause height over the last several decades, based on radiosonde observations, the reanalysis data, and modelling (Highwood et al., 2000; Seidel et al., 2001; Seidel and
230 Randel, 2006; Santer et al., 2003b, a; Meng et al., 2021). The tropopause height derived from the Lauder sonde data shows a significant positive trend of 117±67 m decade$^{-1}$ (at 95% confidence) over the observational period (Figure 10), which is larger than the trend of ∼50-60 m decade$^{-1}$ in the northern hemisphere (20°N-80°N) over 2001-2020 based on radiosonde data in a recent study (Meng et al., 2021). If the tropopause anomaly is not de-trended, the correlation coefficients between the tropopause anomaly and the ozone anomaly are highly anti-correlated with a correlation coefficients of −0.94 at the 9-12 km
layer and −0.95 at the 12-15 km layer. However, with a de-trended tropopause height, the correlation coefficients are −0.71 and −0.73 respectively. This indicates that the negative contribution to the ozone trend in the lower stratosphere (between ∼9 to 15 km) can largely be projected on the significant increase in tropopause height.

The observed tropopause height anomalies at Lauder are also closely correlated with tropospheric temperature (at a correlation coefficient of 0.74) and anti-correlated with stratospheric temperature (-0.76), with significant positive and negative trends
respectively (Figure 10), which are mainly driven by the $CO_2$ increase (Mitchell et al., 1995; Santer et al., 1996). Here, the use of tropopause height as a regressor accounts for the overall dynamical changes, while excluding the effect of inter-dependence of the changes in tropopause height and temperature. Therefore the negative linear trend term accounts for the overall linear effects including the changes in both the stratospheric and tropospheric temperatures.

For the 15-20 km layer, the QBO at 30 hPa also explains a large part of the ozone variability together with tropopause height
and the linear trend term (Figure 8 and Table B1). Above 20 km, the QBO at both 30hPa and 10 hPa, the AOD, together with the tropopause height anomaly explain the ozone variation there, together with a significantly negative linear trend. The correlation between AOD and the Lauder stratospheric ozone is positive, e.g., after the Mt Pinatubo eruption in 1991, despite the potential ozone depletion in the years following a volcanic eruption (Figures 8 and 9). This lack of ozone depletion was attributed to the perturbation of the stratospheric dynamics by the Mt Punatubo eruption that obscured the chemical effect in
the southern extra-tropics (Aquila et al., 2013).





However, in the middle and upper troposphere (6-9 km), the regression function explains the least ozone variations compared to those at levels above and below. Here, although the solar influence is the strongest in relative terms, influences from all other regressors, except the QBO at 10 hPa, contribute non-negligibly to explaining the ozone variations at this level.

In the lower and free troposphere (below 6 km), the sharp decreases in ozone during the early period of the record and the large negative anomalies in 1997/1998 are well reproduced by the regression, as well as the subsequent increases in ozone there. But the large year-to-year variability is less well captured in the free troposphere (Figure 7). This trend transition follows the evolution of EESC, which after a peak in 1997 has been declining since then (Figure 9), and indicates the stratospheric impact on the tropospheric ozone through stratosphere-to-troposphere transport reflecting the effect of stratospheric ozone depletion and recovery. ODS decreases are expected to drive an increase in ozone after the late 1990s, whilst the response to $CO_2$ increases in the troposphere is more complex. For example, the associated increasing humidity would lead to more chemical destruction of ozone in the troposphere, and the increase in temperature may result in more ozone production through $NO_x$-$CH_4$ (and volatile organic compounds) chemistry (e.g., Stevenson et al., 2006; Zeng et al., 2008). Here, relative humidity and surface ozone are anti-correlated (Figure 9). Relative humidity has a large negative impact on surface ozone (Figure 8 and Table B1). We have not considered changes in ozone precursor concentrations and other meteorological parameters in the regression that could substantially impact tropospheric ozone. Indeed, for example, the continuing downward trend in surface ozone after ~2003 cannot be explained by the reduction in ODSs. The regression function we construct here is more suitable to explain the stratospheric ozone changes.

### 3.3.2 Attribution of modelled Lauder ozone changes to ODSs and GHGs

We examine the modelled vertically resolved ozone trends in the vicinity of Lauder (160-180°E and 40-50°S) from the NIWA-UKCA model over the ozone depletion (1987-1999) and recovery (2000-2020) periods separately, as well as the effects of changes in ozone trends due to individual single forcings. Meanwhile, in order to help understand Lauder ozone changes in a global context, we also show the modelled zonal mean ozone trends covering all latitude bands, and the changes that are attributable to ODSs and GHGs in Appendix B (Figures B2 and B3).

Over the ozone depletion (pre-1999) period, the ozone trends at Lauder (Figure 11) are significantly negative (at the 95% confidence level) throughout the height range covered by the sondes, and the magnitude maximizes at $\sim -7\%$/decade at around ~14 km. The modelled trends over this period are broadly in agreement with the Lauder observations (Figure 4). The modelled Lauder ozone trends over this period are attributable to increases in ODS, methane, $N_2O$, and $CO_2$ (Figure 11; the uncertainties of these contributions are displayed separately in Figure B1). The increase in ODSs contributed significantly to the negative ozone trend in the lower stratosphere (~13-25 km), which is the result of the ozone depletion at SH mid-latitudes (Figure B2). The $N_2O$ increase also contribute moderately to the negative ozone trends between ~13-25 km over Lauder but the effect is not statistically significant at 95% confidence (Figure B1). In contrast, the $N_2O$ increase leads to ozone increase in the upper troposphere (5-13 km) as a result of the self-healing effect which was explained by Morgenstern et al. (2018) using the same set of model simulations as used here. The increase in methane during this period (1987-1999) has a considerable positive impact on ozone trend over Lauder below 25 km which maximises at around 10-12 km (Figure 11) and is statistically significant at



the 95% confidence level below 15 km (Figure B1). The ozone increase caused by the growth of methane is partly due to its reaction with chlorine which leads to reduced ozone depletion especially in the stratospheric polar region, and partly through chemical ozone production in the troposphere (Figure B2). The increasing $CO_2$ (derived from the all-GHG forcing and the separate methane and $N_2O$ forcing experiments) has a relatively large negative contribution to ozone over Lauder below 20 km which maximises at a lower altitude of around 10-12 km, but is only statistically significant at the 95% confidence level between
∼7 and 10 km (Figures 11 and B1).

The stratospheric equivalent chlorine attained its maximum in the late 1990s and has been declining since. Consequently, over the period of 2000-2020, the model shows a largely significant positive ozone trend of up to 2% decade$^{-1}$ above ∼23 km in the stratosphere (Figures 11 and B1). In the lower stratosphere, however, a small negative ozone trend of less than 2% in magnitude occurs between 15 and 25 km, and the trend is statistically significant between 17 and 22 km. This simulated
negative trend is about half in magnitude compared to the observed trend at Lauder which covers a larger vertical domain from 8 km to 30 km (Figure 4). In the troposphere below 8 km, the modelled and the observed trends are both positive which are up to ∼2% decade$^{-1}$ in magnitude. In a recent assessment, combined satellite datasets indicate a negative trend over the period of 2000-2020 in the SH mid-latitude (35-60°S) of the lower stratosphere, but multi-model results generally show non-significant positive trends (Godin-Beekmann et al., 2022; WMO, 2022), which is typically associated with a large dynamical variability
in this region. The effects of ODSs, methane, and $N_2O$ on ozone above 15 km are generally small and insignificant, but they become larger and sometimes significantly positive below 15 km (Figure 11). In contrast, the impact of $CO_2$ on ozone at Lauder are negative below 20 km (statistically significant between ∼6-14 km) which maximises at ∼12 km (Figures 11 and B1). The $CO_2$ increase also results in ozone reduction in the tropopause region globally (Figure B3). With declining ODSs, $CO_2$ plays a dominant role in driving ozone trends in the future. The model results support the finding in our regression analysis that the
impact of $CO_2$, reflected in changes of tropopause height (Figures 7 and 8), is an important driver of negative ozone trends in the lower stratosphere in the Lauder ozonesonde record, especially after 2000 while the ODSs are declining (Figure 4). The result here is consistent with previous findings on the response of global ozone changes to ODSs and GHGs using either the CCMI-1 (Morgenstern et al., 2018) or Aerosol and Chemistry Model Intercomparison Project (AerChemMIP) simulations (Zeng et al., 2022).

## 4   Conclusions

We have updated the Lauder ozonesonde timeseries by homogenising the dataset with a series of well-defined correction steps accounting for changes in hardware and operating procedure. We have analyzed this homogenised dataset for height-resolved ozone trends over the 1987-1999 and 2000-2020 periods, characterised by increasing and decreasing trends of total chlorine and bromine, respectively. There are significant differences between the homogenised and the uncorrected data for the pre-1999
period due to these corrections, in which the uncorrected data are low-biased compared to the homogenised data in general. This leads to significantly stronger negative stratospheric ozone trends in the homogenised data compared to the uncorrected data over the 1987-1999 period. The homogenised data typically show negative ozone trends of ∼ −7 to −2% decade$^{-1}$ from





the surface to 30 km with a maximum of $-9\%$ decade$^{-1}$ around 13 km, with significant trends at the 95% confidence above 12
and below 5 km. In both these altitude regions the trends are substantially stronger than trends in the uncorrected data which
are largely insignificant. For the post-2000 period, the homogenisation does not alter ozone trends significantly; both datasets
show significant negative trends in the stratosphere up to $\sim -5\%$ decade$^{-1}$ and small positive trends of up to $+2\%$ decade$^{-1}$
in the troposphere. The post-2000 trends in ozonesonde data are in excellent agreement with trends in co-located FTIR ozone
profiles.

By using a multiple linear regression analysis we have identified the dominant factors driving the Lauder vertically resolved
ozone trends and variations. The regression model consists of independent regressors including solar flux, the state of ENSO,
the QBO at two different altitudes, stratospheric equivalent chlorine, and the aerosol optical depth representing volcanic influ-
ences. A linear term accounts for monotonically changing anthropogenic forcings (led by $CO_2$). Additionally we have included
the detrended tropopause height anomaly, representing the dynamical variability that drives the interannual variability in ozone,
and surface relative humidity that reflects the effect of humidity on near surface ozone, as regressors. We find a persistent neg-
ative stratospheric ozone trend at Lauder represented by the significant negative trends in the linear term of the regression
function. The variation in tropopause height, which anti-correlates with stratospheric but correlates with tropospheric tem-
perature, largely explains the interannual variations in upper tropospheric and lower stratospheric ozone. Significant trends in
tropopause height (positive), the stratospheric temperature (negative), and the tropospheric temperature (positive) measured at
Lauder are consistent with well-established impact of stratospheric circulation changes driven by $CO_2$ increases (e.g., Mitchell
et al., 1995; Butchart et al., 2006). The QBO and AOD indices explain much of the stratospheric ozone variations above 20 km.
In the troposphere, the interannual variations and trends in ozone are less well explained by the regression function. The ODSs
correlate with downward and upwards trends in tropospheric ozone before the late 1990s and after 2000 while ODSs were
increasing and decreasing, respectively. Surface relative humidity explains a substantial amount of surface ozone variability.

We have also used a series of chemistry-climate model single forcing simulations to gauge the impact of changes in GHGs,
including methane, $N_2O$, and indirectly $CO_2$, and ODSs on ozone profiles at Lauder, as well as the zonal mean ozone profiles
covering all latitude bands in a global context. For 1987-1999, simulations show significant negative ozone trends throughout
the vertical domain (up to 35 km), broadly in agreement with observed ozone trends at Lauder during this period. Single
forcing simulations attribute the negative ozone trend to ODS-driven ozone depletion in the SH mid-latitudes and increases
in $CO_2$ which lead to changes in stratospheric circulation and temperature that impact ozone. However this negative impact
on ozone is offset by the positive impact of methane. $N_2O$ plays a smaller role with both negative impacts on ozone above
$\sim 13$ km and positive ones below that level. Over the period of 2000-2020, the model underestimates the significant negative
ozone trend in the lower stratosphere over Lauder. The sensitivity simulations point to a significant negative impact of the $CO_2$
increase on ozone in the upper troposphere and the lower stratosphere, offset by positive impacts from the reduction in ODSs
and increases in methane and $N_2O$. This modelled negative impact from $CO_2$ on ozone through dynamical changes is reflected
in the observed tropopause height increase at Lauder, and this impact will grow if $CO_2$ is continuously increasing in the future.
Therefore, long-term vertically resolved monitoring of ozone is of particular importance to understanding the impact of climate
change on the ozone distribution and vice versa.





*Data availability.* The "uncorrected" Lauder ozonesonde data can be accessed at the World Ozone and Ultraviolet Radiation Data Centre (WOUDC) archive (https://woudc.org/data/explore.php) and at the Network for the Detection of Atmospheric Composition Change
(NDACC) archive (https://www-air.larc.nasa.gov/missions/ndacc/data.html). The homogenised Lauder ozonesonde data can be obtained from the TOAR-II HEGIFTOM Focus Working Group (https://hegiftom.meteo.be/datasets). The Stratospheric Aerosol Optical Depth data was obtained from the NASA Langley Research Center Atmospheric Science Data Center (https://asdc.larc.nasa.gov/).

## Appendix A: Homogenisation of Lauder ozonesonde time series

The corrections that are applied to the Lauder Ozonesonde time series are detailed below. All the corrections are applied on the
360 raw ozone currents. When these cell current were not archived in the early period, they need to be reconstructed from the ozone partial pressure data in the NDACC archive with the available metadata (e.g. pump flow rate, pump temperature, background current, pump efficiency correction table used). Then, correction functions are applied according to those recommended in Smit and the O3S-DQA panel (2012). The effect from each correction is shown in Figure A1.

### A1    Conversion efficiency

The stoichiometry correction was applied for the 1986 data where 2.5 ml instead of 3 ml of cathode solution was used. The EnSci sondes with a 1.0% buffer solution strength over the period of 1994-1996, instead of a 0.5% strength, were also corrected.

### A2    Background current

A consistent background current correction was applied to the Lauder data. If the background current values fall above the mean value + 2 standard deviation ($\sigma$), these values are replaced by the mean value. The mean and corresponding standard
deviations are calculated and applied separately in two periods (i.e., before and after 1996), as the background current values are systematically larger for the period before 1996 and smaller for the period after 1996.

### A3    Pump temperature measurement

Truest pump temperature correction is applied according to Eq. 13 of the O3S-DQA Guidelines (Smit and the O3S-DQA panel, 2012). SPC-4A sondes (until 1989), SPC-5A (from 1989 to 1994), and EnSci sondes (from 1994) were launched in the con-
375 figuration where the pump temperature measurement was made inside the pump. However, the SPC-4A and SPC-5A pump temperature measurements need additional corrections (see Smit and the O3S-DQA panel, 2012).

### A4    Pump flow rate (moistening effect)

Eq. 15 of the O3S-DQA Guidelines was applied to correct the moistening effect of the pump flow rate. There are missing metadata including temperature and humidity of the laboratory before February 2014. The climatological means calculated for
each month are then used for these missing metadata.



## A5 Pump flow efficiency

Eq.22 of the O3S-DQA Guidelines (Smit and the O3S-DQA panel, 2012) was applied using the Pump flow correction factors (CPF) as a function of air pressure (Table 6 of this guideline). These are also applied on the "uncorrected data", as a part of the conversion from the ozone currents to ozone partial pressures. The small change in these correction factors around 1994 is due
to the fact that different correction factors need to be applied for SPC and En-Sci ozonesonde pumps.

## Appendix B: Supplementary figures and table

Figure B1 displays the modelled Lauder ozone trend changes due to ODSs, methane, $N_2O$, and $CO_2$ including the $2\sigma$ uncertainty range. Figures B2 and B3 display the modelled zonal mean ozone trends and the impacts from ODS, combined GHGs, methane, $N_2O$, and derived $CO_2$ for the periods of 1987-1999 and 2000-2020, respectively. Table B1 contains the coefficient
of determination and the regression coefficients from the multiple linear regression analysis.

*Author contributions.* RQ, HS, AG, PS carried out Lauder ozonesonde measurements and processed the data. RVM and DP helped with homogenisation of Lauder ozonesonde data, DS and JR provided FTIR ozone time series, GZ and OM performed model simulations and conducted the statistical analysis. GZ led the writing of the paper with inputs from all authors.

*Competing interests.* The authors declare that they have no competing interests.

*Acknowledgements.* This research was supported by the NZ Government's Strategic Science Investment Fund (SSIF) through the NIWA programmes CACV and CAAC. We acknowledge the contribution of NeSI high-performance computing facilities to the results of this research. NZ's national facilities are provided by the NZ eScience Infrastructure and funded jointly by NeSI's collaborator institutions and through the Ministry of Business, Innovation & Employment's Research Infrastructure programme (https://www.nesi.org.nz).



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





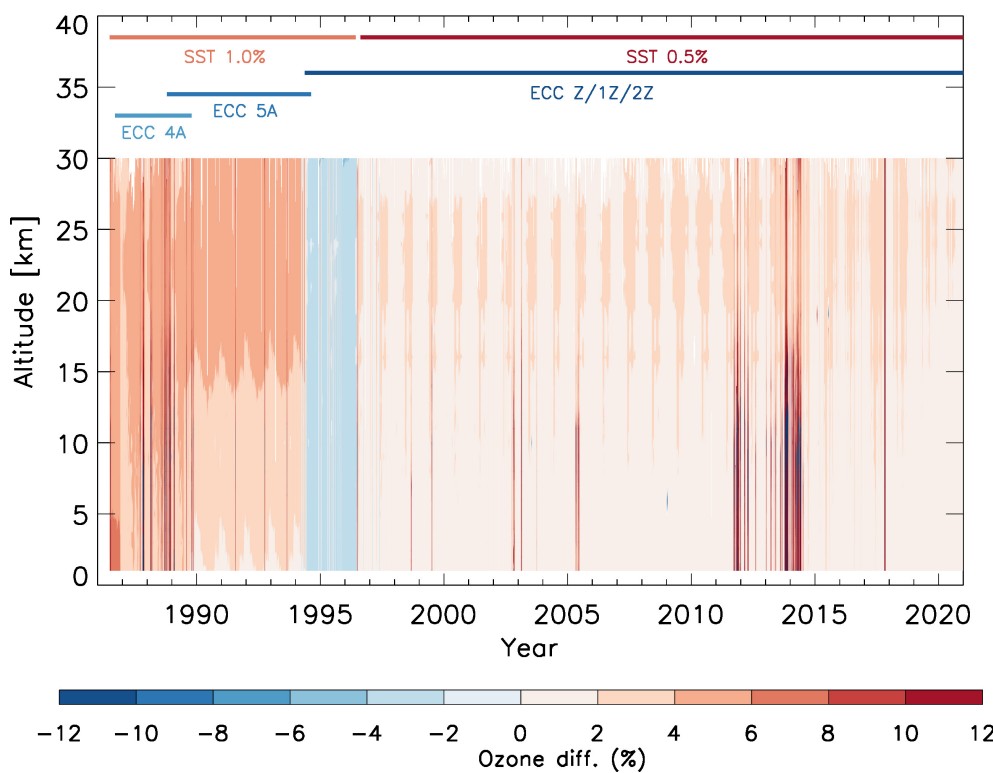

**Figure 1.** Comparison of ozonesonde timeseries before and after homogenisation over 1987-2020 for all flights, in percentage difference between "homogenised" data and "uncorrected" data (i.e., $100 \times (homogenised - uncorrected)/uncorrected$). Also shown are periods indicating changes in the ozonesonde type and the solution used.





**Figure 2.** The homogenised and the uncorrected monthly mean ozone values (ppbv) for different vertical layers over 1987-2020. For displaying purposes, the monthly data are smoothed with a 3 box-car filter.




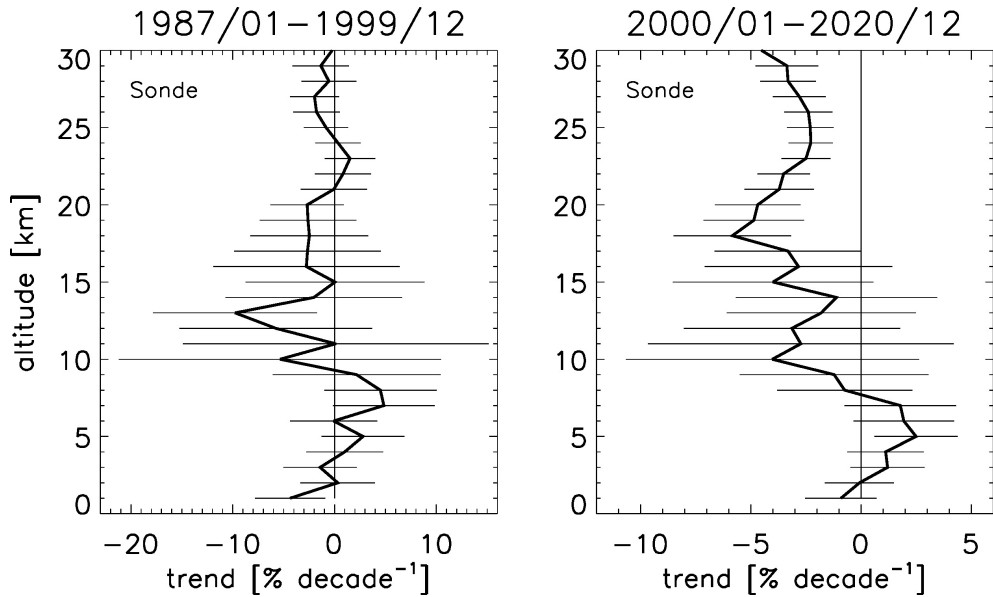

**Figure 3.** Vertically resolved observed trends in monthly mean ozone (uncorrected) and their uncertainties ($\pm 2\sigma$) at Lauder over two periods, i.e., 1987-1999 and 2000-2020.

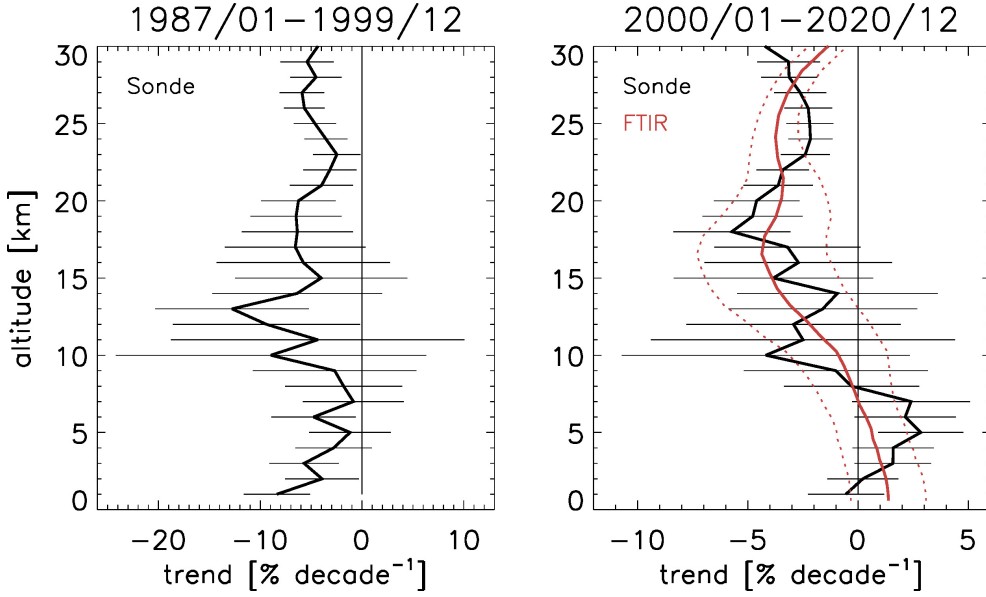

**Figure 4.** Vertically resolved observed trends in monthly mean ozone (homogenised) and their uncertainties ($\pm 2\sigma$) at Lauder over two periods, i.e., 1987-1999 and 2000-2020, from ozonesonde measurements (black), and from FTIR measurements (red, for the 2001-2021 period).





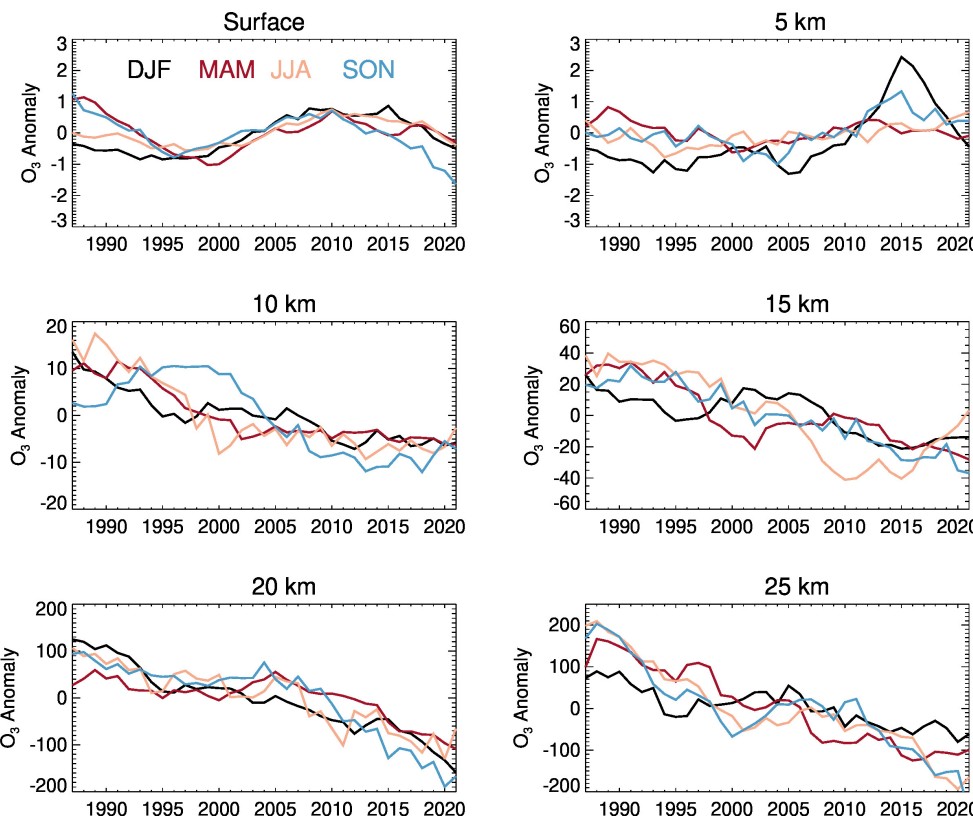

**Figure 5.** Seasonal anomalies in homogenised monthly mean ozone sonde data, averaged for December-January-February (DJF), March-April-may (MAM), June-July-August (JJA), and September-October-December (SON).





**Figure 6.** Standardised time-varying regressors (the tropopause height time series has been de-trened) used in multi-linear regression for Lauder ozonesonde time series between 1987 and 2020. Monthly data are smoothed using a 12-boxcar filter.





**Figure 7.** Observed ozone anomalies (black curves) and regressed ozone anomalies (red curves) at Lauder (1987-2021) for eight vertically averaged layers.







**Figure 8.** Standardised regression coefficients $\pm 2\sigma$ (Standard errors) for eight vertically averaged layers. Values of the regression coefficients and their standard errors are also listed in Table B1.



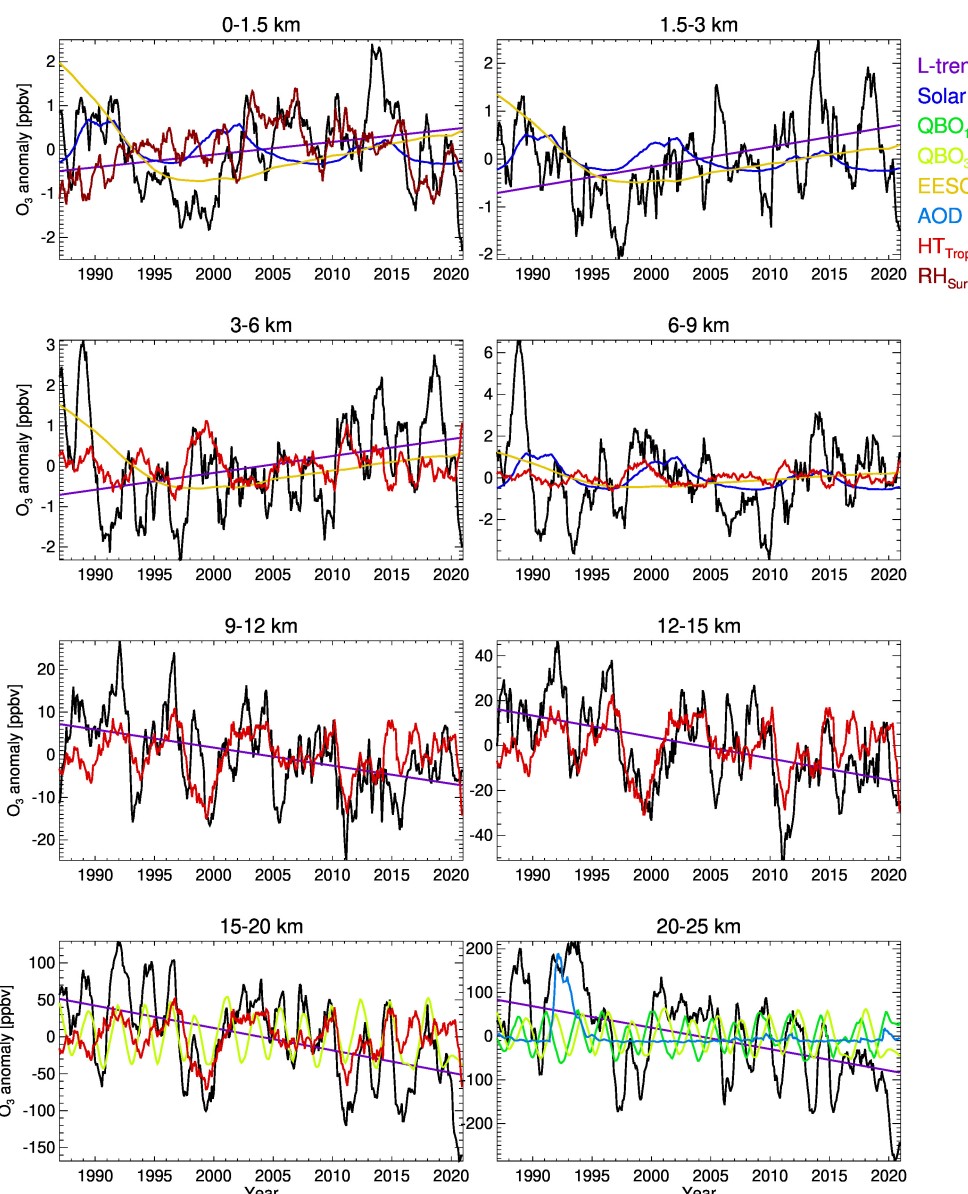

**Figure 9.** Leading regressors (coloured curves) contributing to observed ozone anomalies (black curves) at Lauder (1987-2021) for eight vertically averaged layers.





**Figure 10.** Observed tropospheric (averaged below 8 km) and stratospheric (averaged between 15 km and 20 km) temperature anomalies and the tropopause height anomaly at Lauder. Monthly mean time series are smoothed using a 12-boxcar filter for displaying purposes. Linear trends $\pm 2\sigma$ shown in the plot are calculated based on monthly mean data without smoothing.





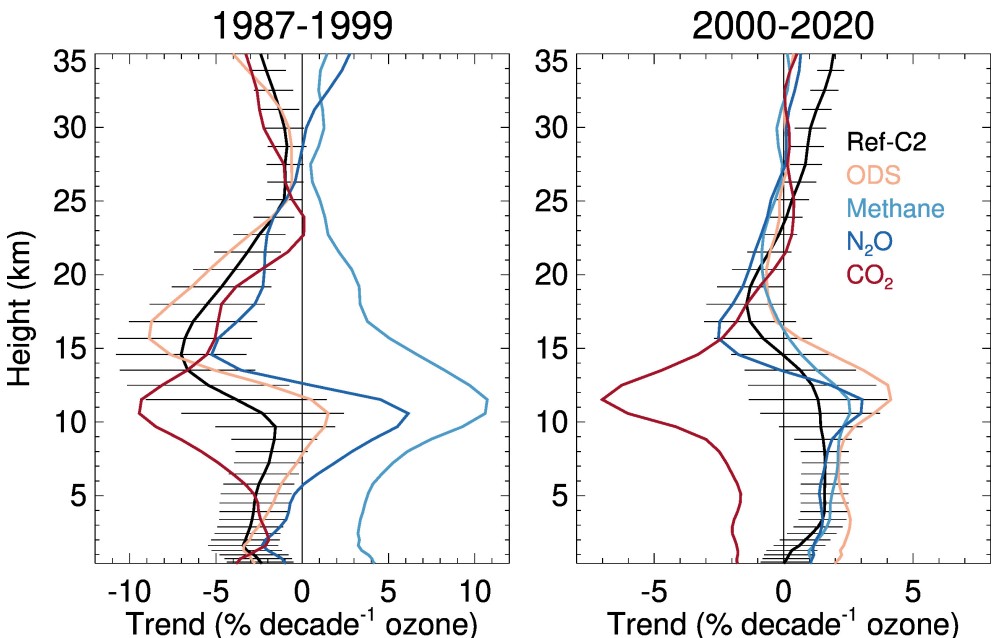

**Figure 11.** Trends in ozone averaged in the area of 160°E-180°E and 40°S-50°S (representing the location of Lauder) for the periods of 1987-1999 (left) and 2000-2020 (right) from the NIWA-UKCA CCMI RefC2 simulation (Ref-C2) with the $2\sigma$ uncertainty range, and the trend changes due to changes in the ozone depleting substances (ODS), methane, nitrous oxide ($N_2O$), and $CO_2$ over the same period.



**Figure A1.** Effect of various corrections on Lauder ozonesonde time series, expressed in percentage changes in ozone.



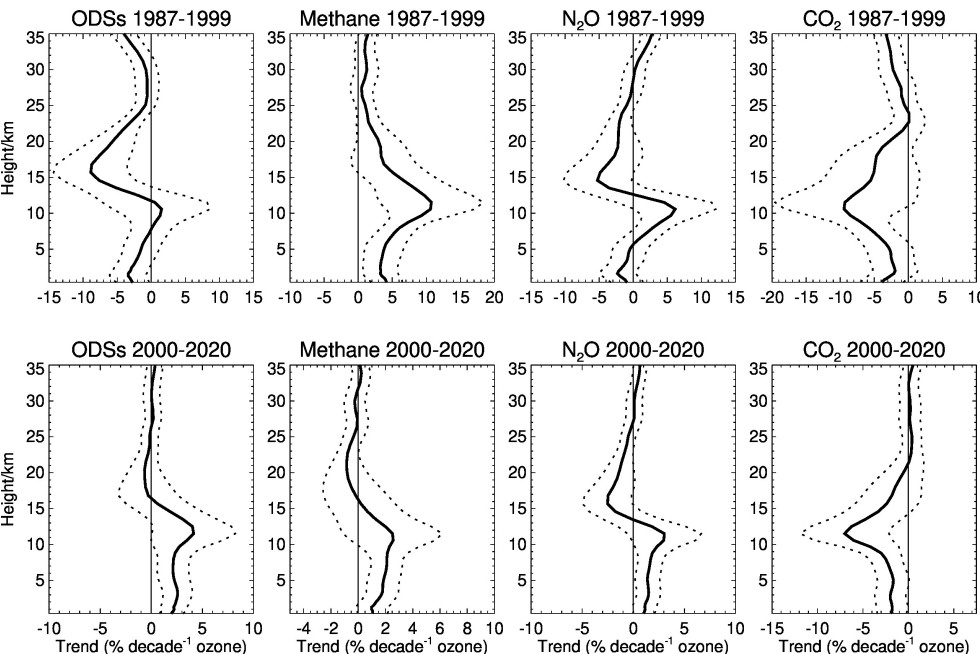

**Figure B1.** Ozone trend changes due to changes in the ozone depleting substances (ODS), methane, nitrous oxide (N$_2$O), and CO$_2$ for the periods of 1987-1999 (up) and 2000-2020 (bottom) simulated in NIWA-UKCA CCMI simulations. The 2$\sigma$ uncertainty range is marked by dotted lines. Trends in ozone are averaged in the area of 160°E-180°E and 40°S-50°S (representing the location of Lauder).

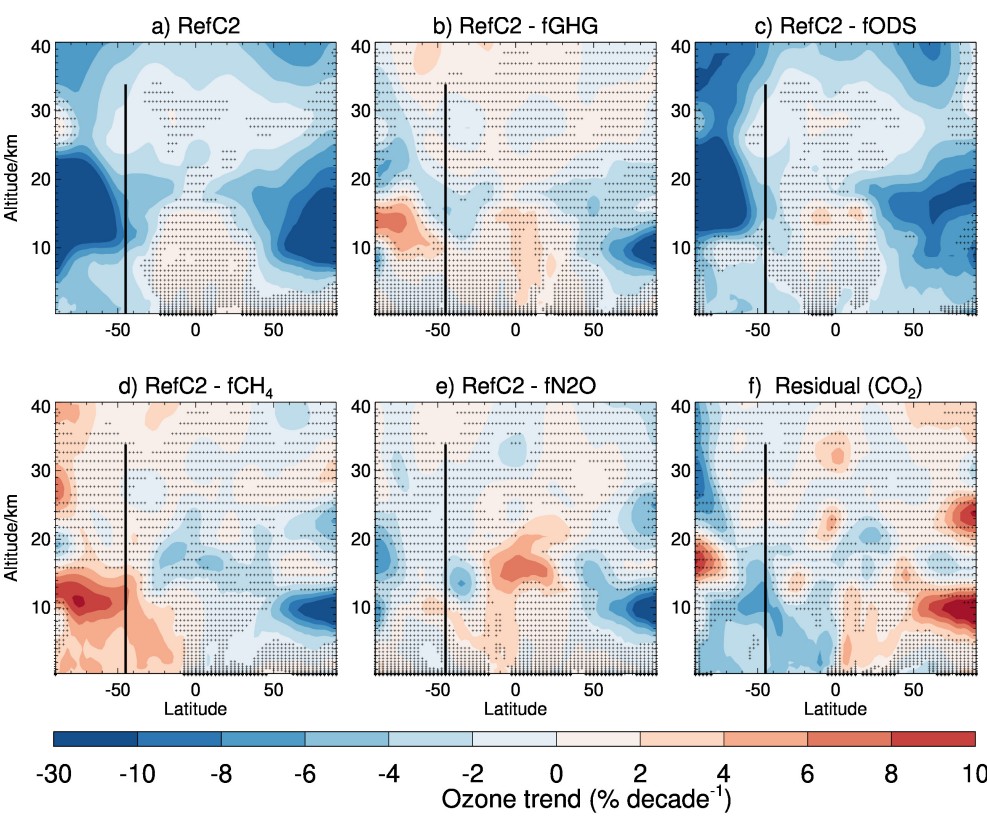

**Figure B2.** Trends in zonal mean ozone between 1987 and 1999 from the NIWA-UKCA CCMI RefC2 simulation (a), and the change in zonal mean ozone trend due to changes in b) greenhouse gases (GHGs), c) ozone depleting substances (EESC), d) methane ($CH_4$), e) nitrous oxide ($N_2O$), and f) $CO_2$ over the same period. Black vertical lines indicate the latitude of Lauder Station.



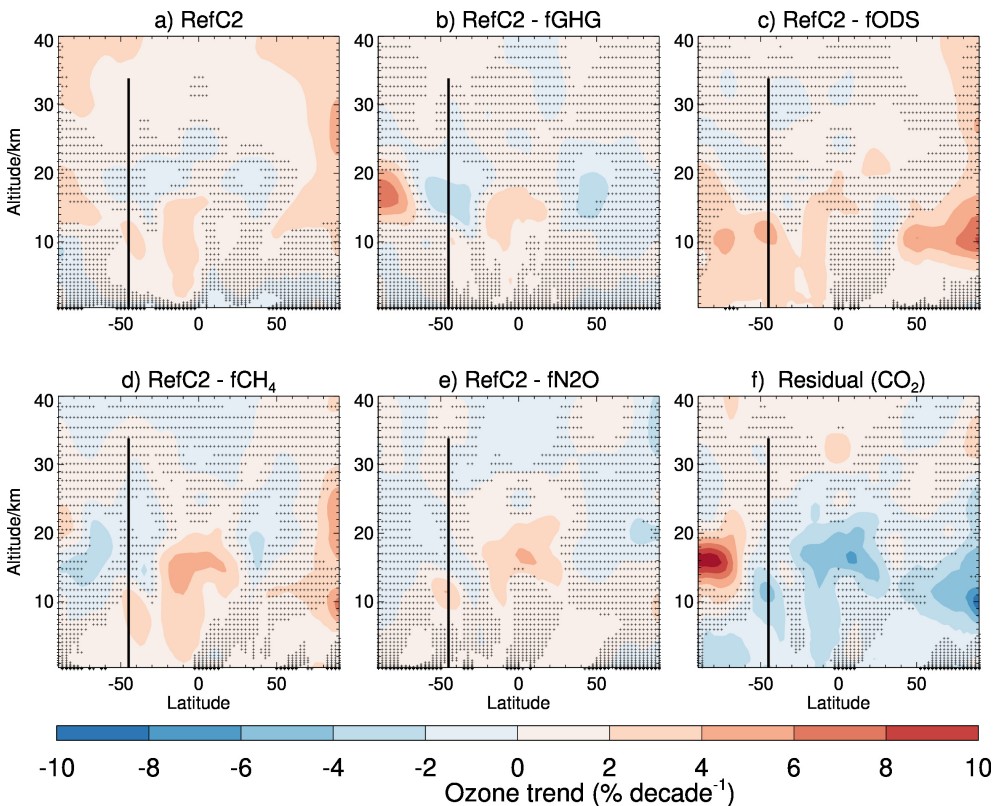

**Figure B3.** As Figure B2, but for the period of 2000-2020.





**Table 1.** Changes in ozonesonde types and solutions.

| | | |
|---|---|---|
| Ozonesonde type changes at Lauder | | |
| Science Pump | ECC 4A | August 1986 to October 1989 |
| (4A/5A/6A) | ECC 5A | 1988 (3), August 1989 to 1995, 1996 (2), 1997 (2) |
| | ECC 6A | 1997 (2) |
| EnSci | ECC 1Z | May 1994 to 2016 |
| (1Z/2Z/Z) | ECC 2Z | 2000 (1), 2001 to present |
| | ECC Z | 2007 (2), 2008 - present |
| Sensing solution changes at Lauder | | |
| KI 1.0% SST | | August 1986 to Jul 1996 (incl. 3 dual flights for comparison) |
| KI 0.5% SST | | August 1996 to present |
| Note: 2.5 ml instead of 3 ml of cathode solution was used in 1986. 1.5 ml of anode solution is always used. | | |

Numbers in brackets indicate the numbers of flights in these conditions.

**Table 2.** Forcings for regression model.

| Variable | Description | Source |
|---|---|---|
| $Solar(t)$ | Monthly mean 10.7 cm solar flux | https://psl.noaa.gov/data/correlation/solar.data |
| $SOI(t)$ | Multivariate ENSO Index Version 2 (MEI.v2) | https://www.psl.noaa.gov/enso/mei |
| $QBO\_10(t)$ | Orthogonalised Singapore winds at 10 hPa | https://acd-ext.gsfc.nasa.gov/Data_services/met/qbo/ |
| $QBO\_30(t)$ | and 30 hPa | QBO_Singapore_Uvals_GSFC.txt |
| $EESC(t)$ | Equivalent Effective Stratospheric Chlorine | RCP6.0 Scenario (WMO, 2011) |
| $AOD$ | Stratospheric Aerosol Optical Depth | NASA/LARC/SD/ASDC (2022) |
| $HT_{Trop}(t)$ | Tropopause height | WMO lapse rate definition (WMO, 1957) |
| $RH_{surf}(t)$ | Relative humidity at the surface | Measured |





**Table B1.** Coefficient of determination, $R^2$, for each altitude band (km), and the standardised regression coefficients $\pm 2\sigma$ (Standard error).

| Height (km) | $R^2$ | Standardised Regression Coefficients ($a_{1-9}$) | | | | | | | | |
|---|---|---|---|---|---|---|---|---|---|---|
| | | L-trend | Solar | SOI | QBO10 | QBO30 | EESC | AOD | $HT_{Trop}$ | $RH_{surf}$ |
| 0-1.5 | 0.50 | 0.28±0.08 | 0.31±0.07 | -0.09±0.08 | -0.01±0.07 | -0.12±0.07 | -0.64±0.08 | -0.01±0.07 | -0.16±0.07 | -0.56±0.08 |
| 1.5-3 | 0.41 | 0.41±0.08 | 0.23±0.07 | -0.12±0.08 | 0.08±0.07 | -0.15±0.07 | -0.44±0.08 | 0.04±0.07 | 0.06±0.07 | -0.13±0.08 |
| 3-6 | 0.43 | 0.41±0.10 | 0.06±0.10 | -0.08±0.10 | 0.16±0.09 | -0.19±0.09 | -0.50±0.11 | -0.06±0.09 | 0.38±0.09 | 0.08±0.11 |
| 6-9 | 0.27 | 0.21±0.19 | 0.52±0.18 | -0.21±0.19 | 0.08±0.16 | -0.20±0.16 | -0.40±0.20 | -0.24±0.17 | 0.30±0.17 | 0.22±0.20 |
| 9-12 | 0.50 | -4.17±0.77 | -0.06±0.73 | -1.62±0.77 | -0.16±0.67 | 0.19±0.67 | -0.38±0.83 | 0.73±0.71 | -5.07±0.70 | 0.63±0.82 |
| 12-15 | 0.61 | -9.38±1.39 | -2.32±1.31 | -0.50±1.38 | -0.08±1.19 | 0.03±1.20 | -2.16±1.48 | 1.56±1.27 | -10.5±1.26 | 0.37±1.47 |
| 15-20 | 0.73 | -29.8±3.84 | -5.17±3.63 | -2.38±3.82 | -3.69±3.30 | -27.5±3.31 | 2.18±4.10 | 9.93±3.52 | -24.2±3.49 | 2.55±4.08 |
| 20-25 | 0.67 | -48.3±7.33 | 16.1±6.94 | -6.51±7.31 | -32.2±6.31 | -32.5±6.34 | 3.36±7.84 | 33.0±6.74 | -18.2±6.67 | -1.91±7.79 |