# Peer review of "Analysis of a newly homogenised ozonesonde dataset from Lauder, New Zealand"

_EGUsphere, 2023_

## Author Comment (AC1)

We thank both reviewers for their encouraging comments and very useful suggestions, which have helped improve the manuscript. We respond to each comment and suggestion that was raised by the reviewers and revised the manuscript accordingly. Our response is highlighted in blue.

We summarise the major changes we made to the revised manuscript as below.

1) In response to the reviewers' concerns, we have modified the MLR model by removing the linear term, using non-detrended tropopause height for regressor, and added stratospheric temperature (averaged between 22 and 30 km) for regression. The stratospheric temperature accounts for the $CO_2$ impact above the UTLS region where the impact of tropopause height change is minimal. As a result, the related texts and discussions have been changed accordingly.

2) We use annual mean ozone anomalies for all trend calculations and for MLR regression to minimise autocorrelation in the data. The trend values are similar with those calculated using the monthly data but with some change in the uncertainty range.

3) We added vertically resolved linear trend in regressed ozone and compare it with the linear trend in observed data (New Figure 4). The trend values are similar, but the uncertainty range is smaller in the MLR trend. There is a systematic underestimation of the MRL trend above 18km indicating that the MLR regressors do not fully capture the observed trend there, although the difference is quite small.

4) We have used all available ensemble numbers of the NIWA-UKCA simulations, in contrast to that only one ensemble number of each simulation was used in the original manuscript. This results in a small change in the modelled trend, but there are no qualitative changes.

5) We have added the contribution of EESC, tropopause height, and the combined tropopause height and stratospheric temperature (to more completely account for the $CO_2$ effect) to the trend in regressed ozone to compare with the model attribution (new Figure 7).

6) We have combined Figures 7 and 9 (new Figure 5), and removed Figures 5, 6 and 8.

7) The subsection headings have changed slightly to better reflect the contents.

[Figure]

**New Figure 4**. Vertically resolved linear trends in regressed ozone (black) and in homogenised observed ozone (red) and their uncertainties (±2σ) at Lauder over two periods, i.e., 1987-1999 and 2000-2020. Data used for linear trend calculations in both cases are annual mean anomalies.

[Figure]

**New Figure 5**. Regressed ozone anomalies (black curves) and Observed ozone anomalies (black dotted curves, homogenised data) at Lauder (1987-2021) for eight vertically averaged layers. Contributions from Leading regressors for each layer are displayed in coloured curves (colour keys in the right of the plot.

[Figure]

**New Figure 7**. Upper panel: Vertically resolved trends in modelled annual mean ozone anomalies averaged in the area of 160E-180E and 40S-50S (representing the location of Lauder) for the periods of 1987-1999 (left) and 2000-2020 (right) from the NIWA-UKCA CCMI RefC2 simulation with the 2σ uncertainty range (black), and the trend changes due to changes in the ozone depleting substances (ODSs), methane, nitrous oxide (N₂O), and CO₂ over the same period. Lower panel: Vertically resolved ozone trends in the predicted ozone by the multiple linear regression (black) and the contribution from EESC (orange) , the tropopause height (red), and the combination of the tropopause height and the stratospheric temperature changes (dotted red)

**Response to reviewers' comments**

Our responses are highlighted in blue.

**Review #1**

The paper describes an updated / homogenized ozone sonde dataset from the Lauder station in New Zealand. It shows results from a trend analysis of the updated sonde dataset, as well as trend results for the region around New Zealand from simulations by the NIWA-UKCA chemistry climate model. The Southern Hemisphere is a region with quite sparse ground-based observations, so Lauder is a very important station and the presented material is, in principle, well suited for publication in ACP.

Overall the manuscript is generally well written, the Figures are clear, and the use of English is generally good.

Nevertheless, I have a number of questions and suggestions that should be addressed before the paper can be accepted for ACP.

We appreciate the reviewer's encouraging comments and address individual comments as below.

In section 2.2 the authors introduce their multiple linear regression model, which includes a single linear trend and EESC as proxies for long-term variations / trends. However, in section 3.2 / Figs. 3 and 4, they show linear trends for two periods (1987 to 1999 and 2000 to 2020). It seems that these trends were calculated with just one linear trend proxy in a simple linear regression, without additional proxies. Is that correct? If so, that needs to be stated very clearly, and the sequence of the (sub)-sections should maybe be reordered. In the current text, this has confused me, and will probably confuse most readers. Alternatively, two linear trends (one over each period), or a trend and change of trend (hockey stick), could be used in the multiple linear regression to be consistent throughout the paper.

The ozone trends shown in Figures 3 and 4 are indeed linear trends calculated for pre-2000 and post-2000 period separately. We have clarified this in the revised version. We have also combined Figure 3 and 4 into new Figure 3.

We have now added the linear trend calculated for the ML regressed homogenised ozone for comparison with that calculated directly from the observed (new Figure 4). They are broadly consistent but the uncertainty for the observed linear trends is slightly larger.

In constructing the MLR model, we use EESC as the regressor which resembles a hockey stick. We then calculated two different linear trends for the two separate periods, to compare with linear trends calculated directly based on the observed data.

The same kind of question applies to the trends for the CCMI simulations. Were these obtained with just a simple linear trend, over the two different periods, or with the full multiple linear regression model?

Indeed, the trends for the CCMI modelled ozone are just simple linear trends for the two separate periods, calculated from the diagnosed annual mean anomalies for each vertical level, the same as in Zeng et al. (2022). We have clarified that in the revised manuscript. We do not use MLR on the modelled data. Instead, we use the sensitivity simulations to attribute the causes to changes in ozone trends.

In line 134, the authors state that "Observations as well as basis functions are smoothed using a 12-month boxcar filter". This is not a usual approach and could affect the derived uncertainties quite significantly, because essentially the number of independent data points is reduced by a factor of 10. How is this accounted for in the uncertainties? How are the uncertainties derived in the first place? How do the authors account / correct for autocorrelation in the residuals ($\epsilon(t)$ in their Eq. 1)? How would the results and uncertainties look without doing this 12-month boxcar? I would assume that much lower values for $R^2$ would be found compared to the quite high values in Table B1. My suggestion is to not use this 12-month boxcar, and go with the standard approach using monthly means without smoothing. In any case, these questions need more analysis and more discussion in the manuscript.

Indeed, the regression using smoothed data has introduced autocorrelation. We now use observed annual mean anomalies in constructing the regression model. In this case, the autocorrelation of the data has been largely removed (as shown by the Durbin-Watson test), but the uncertainties have increased. The results are largely unchanged.

We have decided not to use the monthly mean data as the simple MLR we applied here cannot capture the very noisy monthly signal. We aim to identify the drivers for large-scale interannual variations and trends. We deliberately use the regressors that are known to impact the trends and variability of ozone, e.g., the tropopause height changes, that are not included in the LOTUS regression model. Capturing seasonal variations would require a more complicated MLR with more predictors, like the LOTUS regression model. We hope this approach is satisfactory and believe is more suited to quantify the processes that drive the ozone variation and trend. The linear trend calculated for the MLR predicted ozone is very similar to the linear trend calculated for the observed ozone data (new Figure 4), but with a smaller uncertainty range.

Figures 3 and 4: I suggest to combine both figures and plot the trends before and after homogenization in the same plot, in different colors. As the figure stands now, it is very difficult to see how the homogenization changed the trends (very little after 2000?).

We have combined the two figures as suggested (new Figure 3).

Figure 11: This is a very good and interesting Figure. However, I sorely miss the observed trends here. Please include those in the two panels. How do the vertical profiles of regressed EESC and GHG / overall linear trends from the multiple linear regression of the observed data look like in the two different time periods? How does that compare to the corresponding simulated trends (orange lines, red lines in the Figure)? How do the overall trends compare (to the black lines in the Figure).

We have added those regressed trends in the figure (new Figure 7) and the related discussion in the revised version. It shows that the modelled trends underestimate the regressed observed negative trends in general, especially in the UTLS region where the dynamical change is large and difficult to model. We have added some discussion on the difference between modelled and observed trends.

Line 27: Here you write that SH stratospheric ozone trends are dominated (controlled largely might be a better expression) by Antarctic ozone depletion (which is large and significant and controlled by ODS). Yet in your regression you find hardly any significant impact of ODS / EESC (e.g. in Fig. 8). This is quite a big and important discrepancy. Yet in the later parts of the manuscript, in the conclusions and in the abstract, this discrepancy is hardly mentioned at all, let alone resolved. You do mention negative stratospheric ozone trends, but those seem to be from simple linear regression, not multiple linear regression. So what's going wrong / different with the multiple linear regression? Is the right approach used? I think this needs to be cleared / understood.

We have modified the statement at the beginning of the introduction (in line 27) to "Since the late 1970s, due to the release of man-made ozone depleting substances (ODSs), Southern-Hemisphere

stratospheric O3 changes are mainly characterised by Antarctic ozone depletion leading to negative trends in stratospheric ozone``

In this study, we have identified that, although the most important ozone driver before 2000 in the Southern Hemisphere is ozone depleting substances which cause Antarctic ozone depletion, ODSs do not appear to be the only major factor that drives negative ozone trends over Lauder. $CO_2$ plays an important role in driving the regional ozone trends in the lower stratosphere over Lauder.

We have now added the attribution of regressed ozone trend changes due to EESC and the tropopause height (and the combined tropopause height and middle stratospheric temperature) for comparison in new Figure 7. It shows that ODS contribution to the regressed negative ozone trend is comparable to the $CO_2$ impact in the lower stratosphere over the period 1987-1999. The attribution of the ODS and the $CO_2$-driven impacts in the MLR are broadly consistent with the model results. We have emphasised this finding in the revised version.

Line 67ff: Please indicate if these differences are resolved now. It seems that the Lauder sondes trends don't change very much by the homogenization (Fig. 3, post 2000 trends are almost the same, pre 2000 trends have become more negative). So I would assume that your paper does not change the Godin et al. results, except for the Lauder FTIR data trends which have changed and now fit with the sonde trends. I find this question important, and I would like to see answers, both already here, and also later in the paper, e.g. in the conclusions.

We agree that our trend calculations are within the uncertainty range of Godin-Beekmann et al. and we have clarified this in the revised version where appropriate.

Line 130: What happens when tropopause height is not detrended? I think this should be tried and discussed. If non-detrended tropopause height picks up a GHG induced climate-change related ozone trend, that might be the correct way to do the trend analysis. One could argue that the mechanism underlying short term changes of tropopause height and ozone changes also acts on the long time-scale, because climate-change statistically favors high tropopause conditions. So the acting processes could be the same. There may not be the need for a different process acting on the long time scales. E.g. in the annual cycle you also have a close correlation (on a longer time scale) between high tropopause height and low ozone in the lower stratosphere. While this is somewhat different from the short time-scale processes due to high and low pressure systems, both time scales give similar correlation of high tropopause with low ozone.

We have modified the MRL approach and now use the non-detrended tropopause height. We also removed the linear term in MLR, but added stratospheric temperature which explains to a certain degree the negative trend above 20km.

Line 146: Please explain what forcings are included in RefC2. I assume all forcings.

The RefC2 simulations include all forcings (ODS, GHGs, ozone and aerosol precursors). This has been added in the text.

Lines 147, 150: I think this should be "corresponding fixed forcing simulation" not "corresponding single forcing simulation".

Yes, that's correct. We have modified the text.

Around Lines 186, 197: I think these different trends need to be explained / need a bit more dicussion. If Godin et al. get different trends from the same data, there must be an explanation. What happens if you try Godin et al.s regression? Could the difference come from excluding a few extreme soundings? Generally, differences from previous findings need more explanation. They should not just be mentioned and then ignored. I have also done multiple linear regression (with hockey-stick trends) on HEGIFTOM data from Lauder and find trends very similar to your trends in Fig. 4. This is reassuring. Maybe the Godin at al. Lauder trends were too negative? Anyways, I think this needs a bit more discussion, and maybe should be mentioned in conclusions and abstract as well. Of course it needs to be worded appropriately.

We agree with the reviewer that we need to understand the cause of the discrepancy. A rationale for this study is that we verify the trend by calculating the linear trend on the observed time series. As mentioned in the manuscript, excluding some outliers would not change the trend much but would only reduce the uncertainty slightly. We then calculate the linear trend of the regressed observed ozone and find that the difference between them is quite small, especially below 18km where the largest difference occurs between our calculation and Godin-Beekmann's. However, our trends are within the uncertainty range of Godin-Beekmann et al. We have clarified this in the revised version. The observed linear trend and the MRL linear trend in our study also show some small differences in the stratosphere. The reason might be that there is some process missing in the regression, for instance, the recent Australian bush fires that affect the stratospheric ozone but are not considered in the regression. Therefore, it is important to verify the trend calculation with different approaches.

Line 237: As mentioned above, this really begs the question what happens with non detrended tropopause height in the regression.

We have now modified the regression model to use non detrended tropopause height and revised the related discussion.

Line 242: I am not sure about this linear trend term. It seems like a very unspecific overall collector of various things, picking up a confusing mix of ODS-related, GHG-related and other changes. I think there should be better proxies, e.g. hockey stick, two linear trends, ... For me, the insignificant ozone changes picked up by the EESC term are a warning sign. If, according to the regression, ODS changes had no impact on stratospheric ozone at Lauder, I very much wonder if we can trust this regression.

We have now removed the linear term in the MLR model and used non-detrended tropopause height as regressor. We have also added the stratospheric temperature as a regressor to additionally account for the impact of CO2 change in the stratosphere not covered by the impact of tropopause height changes.

The small regression coefficient representing the EESC impact is for the whole period in regression, which could obscure its more significant impact for the pre-1999 period. As shown in new Figure 7, the

impact of ODS on stratospheric ozone trend at Lauder is not negligible in the pre-1999 period, especially in the lower stratosphere in the MLR attribution (Figure 7 lower panel); this is consistent with the model attribution (Figure 7 upper panel). We have explained this in the revised version.

Around line 258: This is a weird argument. We see EESC effects in the troposphere, but we don't see them in the stratosphere, where they should be coming from? Can that be resolved? Might different proxies in the regression help (as suggested above)?

As stated above, the impact of ODS on ozone at Lauder is more important in the pre-1999 period, shown in both the MLR and model attributions (new Figure 7). The small regression coefficient due to EESC in the stratosphere is perhaps obscured by the large impact of $CO_2$-driven dynamical changes throughout the whole observational period, not just for the pre-1999. The post-2000 period sees the impact of EESC dropping considerably therefore the small regression coefficient. However, we do calculate the trend in two separate periods to account for the effect of the EESC's transition from increasing to declining. We hope that this explanation is satisfactory. We have revised the text.

Consequently, the EESC impacts tropospheric ozone at Lauder through transport. Model simulations (Figures B2 (d) and B3 (d)) show that tropospheric ozone at SH mid-latitudes can be influenced by polar ozone changes in the lower stratosphere. Our model simulations also capture the impact of ODS on tropospheric ozone trend in both the pre-1999 and post-2000 periods (new Figure 7).

Around line 320 (and in several other places). Are these simple single linear trends? Or are they from the multiple linear regression? This keeps me confused throughout the manuscript. I would much prefer to just have one type of regression, or a much better explanation of what was done, and why there might have been two approaches.

They are simple linear trends. We have modified the text to make it clear which of the two approaches (LR and MLR) we have used. We have added the MLR trend in the revised version for comparison – they are quite similar. This means that the MLR captures the observed trend well and we have confidence in that it can be used to explain what drive the observed trend.

After line 339: As also mentioned above for Fig. 11, I think there needs to be more discussion about how the observed and simulated trends fit together, or do not fit together.

We have added discussion on the new Figure 7. Although the model underestimates the observed (regressed) trend especially in the lower stratosphere, the roles of ODS and $CO_2$ on ozone are consistent between the model simulation and the regression.

**Review #2**

In the manuscript "Analysis of a newly homogenised ozonesonde dataset from Lauder, New Zealand", Guang Zeng et al. derive long-term MLR trend estimations from the ozone sonde dataset of Lauder. The dataset, recently homogenised in the frame of HEGIFTOM, shows negative pre-1999 trends in better agreement with the previous literature. Post-2000 trends are shown to be significantly negative in the stratosphere and positive in the troposphere, and are in very good agreement with trends derived from a co-located FTIR instrument. The analysis by MLR imputes the negative post-2000 stratospheric trend to anthropogenic forcing led by CO2 related to positive trends in tropopause height and tropospheric temperature and negative trend in stratospheric temperature. CCM simulations from NIWA-UKCA attribute the negative pre-1999 trends not only to the ODS increase but also to a GHG increase with opposing impacts of CH4 and CO2. For the post-2000 period, the CCM analysis assess the role of the dynamical changes driven by CO2 on the negative lower stratospheric trends.

The manuscript fits well within the scope of ACP and is of high scientific quality. It is generally well written despite some very long sentences which do not read well. The results are well presented. The homogenization of the ozone sonde dataset succeeded in improving the pre-1999 trend agreement with other observation techniques and with the literature.

The detailed analysis of the MLR results is a very good contribution to the understanding of the underlying issues of stratospheric and tropospheric pre- and post-2000 trends. The results derived from CCM analysis and derived from MLR on observational dataset enhance the role of CO2-driven dynamical changes in the lower stratospheric trend which represents a significant step towards understanding trends in this atmospheric region.

I list below general and specific comments. General comments are questions and remarks which need clarifications before the paper can be accepted for ACP. Specific comments are minor revisions which may help to improve the readability of the manuscript.

We thank the reviewer for his/her encouraging comments, addressed below.

General Comments

Line 64-74:

The same ozone sonde dataset is used in Godin-Beckmann et al. 2022 and in the present study. The authors say the trend values of Godin-Beckmann et al. to be "exceedingly large" (line 69). However, the present study trends are within the Godin-Beckmann et al. uncertainties and respectively, except for the 25km value. The authors should comment on these differences.

We agree with the reviewer's assessment that the trends calculated in this study are broadly within the range of uncertainty by Godin-Beekmann et al. We have clarified this in the revised version. The different trend calculations (e.g., the linear trend, MLR trend, and LOTUS trend) may have led to some differences. Differences in the regressor selection could lead to some differences in the regressed ozone values. The purpose here is to calculate the simple linear trend based on observed data, although the uncertainty in

observed trend is often larger than the regressed trend, as demonstrated in our revised version in which the observed linear trend and the MLR trend are compared.

Why are the uncertainties of both studies so different? Is the residuals autocorrelation taken into account in the present study? What is the impact of using EESC as an additional explanatory variable on the trend values of the present study?

Indeed, there is autocorrelation in our monthly data used for linear trend and MLR model. In the revised version we use annually averaged ozone anomalies for the trend calculation and the MLR model construction, which largely removed the autocorrelation. As a result, the uncertainty of the trend becomes slightly larger. Our trend uncertainties sit within that by Godin-Beekmann et al.

The EESC explains the tropospheric more than the stratospheric ozone trend. This might be due to that its impact on stratospheric ozone is obscured by the larger impact of dynamical changes (due to $CO_2$) on stratospheric ozone. We have discussed this in the revised version (Sect. 3.4).

Line 74:

"which has been updated from the dataset used in Godin-Beckmann et al. (2022)". Could you please comment already here (instead of Line 194) on the FTIR dataset update as the trends reported in Godin-Beckmann et al. are very different from the present study?

We have revised the text to add comments on updated FTIR retrieval here.

Line 77:

"into the near future": I cannot not see any mention of post-2022 results in the manuscript. What do you mean?

This sentence has been removed in the revised version.

Line 89:

Table1 indicates only 3 dual flights to evaluate the effects of the sensing solution change. Is the transfer function/correction factor derived from these 3 dual flights or is a general transfer function used?

A general transfer function was used.

Line 122:

« Surface humidity are measured » : please replace by: « Surface humidity is measured »

Is this RH?

"Surface humidity are measured by the radiosonde that has a humidity sensor.": please be more specific.

Yes, it is "relative humidity". We have added this information and made the correction.

Line 126:

Why is QBO10 used for the whole altitude range? Would QBO50 have been more appropriate in the troposphere?

We use the same regressors for all levels. We have tested using the QBO50 index but it makes no difference versus using the QBO30 index.

Line 130:

"Fig 6." is the first mention of a figure in the manuscript, in that case, this should be Fig 1.

I suggest to remove this figure or to move it to Appendix B.

We have removed this figure from the revised version.

Line 140:

"coarse resolution » : do you mean spatial or temporal resolution? Please be more specific.

It refers to spatial resolution. "spatial" has been added before "resolution".

Line 177:

"marked differences": the pre-1999 trends values are different but similar within their uncertainties. The marked difference lies in the significance of the trends. This should be made clear. Furthermore, the differences would be clearer if both trend profiles (uncorrected and homogenised) were represented on a single figure.

Indeed, that's the case. We have now combined the original Figures 3 and 4 into new Figure 3 and replace "marked" with "systematic".

Line 184:

"Figure 3 and 4": I suggest to merge the information into one single figure with 2 panels (pre-1999 and post-2000) to make comparisons easier.

We have combined these two figures into one and modified the text accordingly.

Line 187:

same comment as for Lines 64-74. Please explain why the trends values are different in both studies. The role of the additional proxies used is non negligeable. For instance, the significative contributions of the EESC or HTtropo proxies (Table B1) are influencing the remaining linear trend value when compared to a MLR not considering these proxies.

Originally the MLR model contained terms that had not been detrended. This implies that the explicit trend term differs from what would result from a straight linear fit calculation. However, the MLR model has now been changed so the linear trend term no longer exits, and the HT_trop term is not de-trended.

Line 195:

"updated version": See comment for Line 74.

We have revised the manuscript as suggested.

Line 197:

The agreement between ozone sonde and FTIR trends is good however trends of the present study stay negative and significant above 15 km while WMO 2022 shows positive but non significant trends. A reason or a tentative explanation should be given.

Indeed, the combined satellite data show slightly negative trends in the lowermost stratosphere averaged over 35S-60S but with a very large uncertainty. Both CCMI-1 and CMIP6 models show slightly positive trend but none are significant. Lauder's ozonesonde and FTIR data both show significant negative trends. Globally, the role of $CO_2$ in driving the negative ozone trend in the lowermost stratosphere is robust, especially in the tropics to the mid-latitudes (shown in the model attributions on the global scale: Figures B2 and B3). There are also large dynamical variations in this region which may obscure and reduce the significance of the trend.

Line 200-207:

At that point, I cannot find a reason for these considerations on the seasonal variation.

I suggest to remove this section unless it is used in the explanation of the trends estimation.

We have removed this section and the figure in the revised version.

Line 213:

As said in comment of Line 130, Figure 6 could be removed. If kept and moved in Appendix B, please replace "de-trened" by "de-trended" and "12-boxcar" by "12-month boxcar".

We have removed this figure.

Line 210-217 and Figure 7 8,9 and table B1:

Separate and redundant information is spread over 3 figures and 1 table, I find this difficult to handle. I would suggest the following simplification:

Information on Fig 7 is given in Fig 9 except for the R2 value and the regressed timeseries. You could remove Fig 7 and keep only Fig 9 with the observed timeseries in dashed, the regressed timeseries in black and with the R2 information in the respective subplot titles.

We have combined Figures 7 and 9 into the new figure 5 and moved Table B1 to the main text (Table 3).

Figure 8 and table B1:

The information is redundant and Figure 8 could be replaced by a highlighting of the major contributor(s) in table B1. Table B1 should then be part of the manuscript and not of the Appendix B.

Table B1 is now new Table 3 with the major contributions being highlighted in bold. We have removed Figure 8.

Line 223:

"The downward trend in the stratospheric ozone is clearly explained by the significant negative linear trend that represents all quasi-linear, monotonic drivers of change"

Please add which drivers and that you will discuss these below.

Why can these drivers not be used as proxies but have to be treated apart?

We have now modified the MLR and removed the linear term. As suggested, we now use non-detrended tropopause height as a regressor. We have also added stratospheric temperature (averaged between 22-30km) as a regressor to explain the negative ozone trend in the stratosphere (the impact is most significant at above 20km). We have revised the text to accommodate the modification of the MLR analysis.

Why is the contribution of EESC negligeable here?

The contribution of EESC is most significant in the polar lower stratospheric region. In the SH mid-latitude, as we show here, the CO2-driven dynamical changes play a more significant role than the ODSs. Also the effects of ODSs and GHGs are coupled, and it is not easy to separate them in a regression that does not identify causes. From our model results, the role of ODSs is significant in the pre-1999 period (new Figure 7). Discussion on this has been added (e.g., Sect 3.4)

Can you exclude this negative trend to be due (evt partly) to a drift or step in the timeseries which has not been considered or corrected by the homogenisation?

We cannot 100% exclude that there might be drift or step in the time series. However, the study by Bjorkland et al. (2023) (https://doi.org/10.5194/egusphere-2023-2668) shows that there is no significant drift in the stratosphere between ozonesonde and FTIR ozone partial columns at Lauder.

To test, we calculated the total ozone column from integrating the partial columns below 50hPa (which is usually below the balloon's burst point) and an ozone climatology above 50 hPa. We have compared this integrated column with Dobson TCO at Lauder (see the figure below). Although there are some differences between the sonde and Dobson TCO, the data after 2000 does not show visible trend difference between these two datasets, apart from some offsets. The difference between the datasets before 2000 is relatively minor compared to the large change in TCO over this period. We need further investigation on this but is outside the scope of this study.

[Figure]

Line 231:

How is this trend estimated?

Meng et al. trend values are estimated "on the natural variability–removed time series" (volcanic eruption, ENSO and QBO). Are you estimating the trends with a similar model as Meng et al. or with your eq.1? Please comment.

The trend in tropopause height is calculated as simple linear trend of the annual mean anomaly of the tropopause height.

Line 236-7:

"This indicates that the negative contribution to the ozone trend in the lower stratosphere (between ~9 to 15 km) can largely be projected on the significant increase in tropopause height»

Figure 9 shows that de-trended HTtropo is a significant proxy for explaining the ozone variation between 9 and 15 km height. Instead of making considerations about the correlation between de-trend and non de-trended HTtropo and ozone, would it be possible to use non-detrended HTtropo as a proxy? If not, why?

We have modified regression model and now use the non-detrended tropopause height as the regressor and have modified the text accordingly.

Line 267:

"The regression function we construct here is more suitable to explain the stratospheric ozone changes." Does it mean that the trend values estimated below 6 km are not reliable?

The trend in tropospheric ozone is well captured by the regression model (Figure 4) but the interannual variation is less well captured in the free troposphere (Figure 5). We have clarified this.

Line 269-270:

How are the modelled ozone trends estimated? What do you mean by "separately"? If you apply a ILT multi-linear regression on simulated ozone values, please describe the MLR used in that case. If the model directly outputs the trends, please clarify.

The model outputs ozone values in units of mixing ratio. We mean that we calculate trends in the modelled data separately for the two periods (i.e., pre-1999 and post-2000). We have not applied the MLR method to modelled ozone, rather, the modelled ozone trend was calculated as simple linear trend for the two period. We have clarified this in the text.

Line 272:

Please specify here that/if the attribution of the changes in modelled O3 is done as in Zeng et al 2022 and Morgenstern et al. 2018.

Morgenstern et al. 2018 and Zeng et al. 2022 computed the ozone response to changes in each forcing and performed linear regression at each grid point. Then the linear regression coefficient was used as the measure of ozone response to that forcing. In addition, Zeng et al., 2022 also calculated the trend changes due to each forcing in attributing regional ozone changes. We added "the same approach as in Zeng et al. (2022)." After "individual forcings".

Line 275:

" are broadly in agreement with the Lauder observations (Figure 4)." I suggest to add the trend values estimated on the ozone sonde dataset in Figure 11.

We have now added the regressed trend and the contributions from ODSs, HT_Trop, and the combined HT_trop and T_strat  to the new Figure 7.

Line 318:

"with a maximum of −9% decade−1 around 13 km": Figure 4 shows a maximum of -12%/dec at 13km.

"with significant trends at the 95% confidence above 12 and below 5 km. » Please adjust to values shown on Figure 4.

We have adjusted these numbers to the values shown in revised figures.

Specific Comments

Line 5-7:

The sentence doesn't read well. It's too long and contains 3 brackets. Please rephrase.

We have substantially modified the abstract to reflect changes made in the revised version.

Line 10-13:

Please, make 2 sentences from this one.

Revised

Line 19:

"…but clearly shows …" : Please replace by "… , it clearly shows …"

Changed

Line 20:

"have had an increasingly important role": please replace by "have played an increasingly important role"

Revised

Line 20:

"in this region": do you mean in the lower stratosphere or in New Zealand? Please specify.

We refer to the Lauder region - revised.

Line 23:

"and the radiation budget" : please replace by "and in the radiation budget"

Adopted

Line 39:

"over the period 2000-2020, but such observed trends are". Please replace by "over the period 2000-2020. Such observed trends are"

Adopted

Line 42-45:

please make 2 sentences.

Revised

Line 48:

"attempts of attribution using the models": Please replace by "any attempt at attribution using models"

"well-positioned » please replace by "well-suited"

Adopted

Line 58:

Please remove the ) after NDACC

Done

Line 60:

"Any heterogeneities the data have": please replace by. "Any heterogeneities in the dataset »

Adopted

Line 64:

Make a separate sentence with "although we only take the data from January 1987 to December 2020 for analysis here."

Revised

Line 79:

« In the next section », please replace by "In section 2"

Changed to "Sect. 2"

Line79-80:

"section", "Sect.": please make it uniform in the manuscript.

We have adopted "Sect."

Line 98:

"For example » : please replace by "For instance"

Done

"is needed for the change of sensing solution because there was a 2-year period when the EnSci ECCs started to be used, but with the 1% solution, rather than the 0.5% solution which has become the recommendation for the EnSci ECCs." is difficult to read. You could replace that sentence with:

"a transfer function is applied to the data after the change in sensing solution type from 1% to 0.5% KI following the O3S-DQA recommendation"

Btw, is the transfer function applied to the data before or after the change of sensing solution?

A general transfer function is used to correct the profiles that used the 1% KI solution instead of the recommended 0.5% for the EnSci ECCs flights during the period of 1994-1996. We have revised the text to make this clearer.

Line 100:

"re-process » : please replace by "re-processing"

Adopted

Line 105-110:

this paragraph does not read well and need to be rephrased.

For instance: "Both homogenised and the uncorrected datasets have been post-processed for trend calculations and the regression analysis (in the case of homogenised data)."

Shoud be: "Both homogenised and uncorrected datasets have been post-processed for trend calculations and for regression analysis."

What do you mean by "(in the case of homogenised data)"?

We use the homogenised data for MLR analysis.

We have revised this paragraph. It now reads "*In this study, we include a total of 1958 flights between August and June 2021, which the data have been homogenised. Both homogenised and uncorrected dataset have been post-processed for linear trend calculations. The homogenised datasets are used in the MLR analysis. Linear piecewise regression was applied to interpolate the original ozone profiles from the surface to 30~km at a 1~km vertical resolution. We then exclude some extreme ozone values, identified as the values that are outside the 3 standard deviation range, to create monthly means by averaging the data available for that month at each re-gridded vertical level.*''

Line 118:

Please add 3 "the": The tropopause height (HTTrop), the surface relative humidity (RHsurf ), the aerosol optical depth (AOD),

Done

Line 123:

Please replace "WMO (1957)" by "(WMO, 1957)"

Done

Line 124:

"using the co-measured temperature data of each ozonesonde flight": please replace by "from the temperature measured by the radiosonde during each ozonesonde flight"

Adopted

Line 129:

"normalized to vanishing means and unit standard deviation": do you mean "standardized"?

Yes, that's correct. Now we have added "standardized".

Line 152-154:

not clear, I can see redundant information in the same sentence. Please rephrase.

Do you simply mean that the impact of CO2 equals the impact of combined GHGs minus the impact of CH4 and NO2?

Yes, that's correct. We have rephrased this sentence as "*However, no simulation was performed to directly assess the impact of CO2 within CCMI-1; instead, it will be assessed by subtracting the impacts of methane and $N_2O$ from the impact of the combined GHGs.*"

Line 160:

I would use "as measured" instead of "uncorrected" (first mention is on Line 93).

Thanks very much for the suggestion. After some consideration, we have decided to stick with the wording "uncorrected" in contrast to "homogenised".

Line 162:

"in the vertical to 1 km grid using piecewise linear regression for each profile.": please replace by: "to a 1 km vertical grid using piecewise linear regression."

Adopted

Line 167:

"The effect of changes to the concentration of the KI solution on the conversion efficiency"

Please replace by:

"The effect of the changes of the KI solution concentration on the conversion efficiency"

Adopted

Line 169:

Please move "The correction procedure and the impact of each correction are described in more detail in Appendix A." before the reference to Figure A1(3).

Done

Line172:

"outliers where ozone is outside the 3 standard deviation": please replace by: "outliers defined for ozone being outside the 3 standard deviation"

Adopted

Line 202:

"Figure (5)": please replace by "Figure 5"

This paragraph has been removed.

Line 205:

"Some slight differences between seasons are below 5 km": please add "visible" after "are"

This paragraph has been removed.

Line 219-220:

", with R2 ranging from 0.27 to 0.49 in the troposphere and 0.50 to 0.73 in the stratosphere, implying that the stratospheric ozone variations and trends are better explained by the MLR model than tropospheric features.", I would say:

". With R2 values ranging from 0.27 to 0.49 in the troposphere and 0.50 to 0.73 in the stratosphere, the stratospheric ozone variations and trends are better explained by the MLR model than tropospheric features"

Changes made as suggested. There are slight changes in R2 values as the result of modified MLR.

Line 228:

Please remove "the" before "cooling"

Done

Line 229:

Please remove "the" before "reanalysis"

Done

Line 234:

"at the 9-12 km layer": please replace by:  "in the 9-12 km layer"

Done

Line 235:

"at the 12-15 km layer": please replace by:  "in the 12-15 km layer"

Done

Line 238:

Replace "at a correlation"  by "with a correlation"

This paragraph has been modified to accommodate the revised MLR analysis.

Line 245:

Replace the 2nd "together with" by "and"

Done

Line 256:

"This trend transition follows the evolution of EESC, which after a peak in 1997 has been declining since then (Figure 9), and indicates the stratospheric impact on the tropospheric ozone through stratosphere-to-troposphere transport reflecting the effect of stratospheric ozone depletion and recovery." Too much

information for one single sentence. What is "reflecting the effect of stratospheric ozone depletion and recovery " the EESC evolution or the stratospheric impact or the transport…? Please rephrase.

*We have rephrased this sentence as "This interannual variation in tropospheric ozone coincides with the evolution of EESCs which increases since the late 1980s before declining after 1997. This indicates that the impact of stratospheric ozone changes due to changes in ODS could impact tropospheric ozone through transport."*

Line 258:

Please assign a letter or a number to the panels and refer to them as Figure 9 (a) and so on.

*We have updated this figure (new Figure 7) as suggested.*

Line 260 and 265:

Please replace "for example" by "for instance"

*Done*

Line 271:

Please remove "single" between "individual" and "forcings".

*Done*

Line 312:

Please replace "height-resolved" by "vertically resolved"

*Done*

Line 320:

"In both these altitude regions the trends are substantially stronger than trends in the uncorrected data which are largely insignificant ». Trends "in" altitude regions are compared to trends "in" uncorrected data. Please rephrase.

*We have removed this sentence in the revised version.*

Line 326:

Please replace "altitudes" by "pressure levels"

Done

Line 333:

Please add "the" before "tropospheric height"

Done

Line 340:

Please add "on" before "the zonal mean ozone profiles"

Done

Line 343:

"and increases in CO2 which lead": please replace by "and to increase in CO2 which leads"

Adopted

Line 356:

Please remove "The Stratospheric Aerosol Optical Depth data was obtained from the NASA Langley Research Center Atmospheric Science Data Center (https://asdc.larc.nasa.gov/)" from the Acknowledgements but add "(https://asdc.larc.nasa.gov/) in Table 2.

Done

Line 291:

Please replace "attained" by "reached"

Done

Line 302:

Please replace "are negative" by "is negative" and "which maximises" by "and maximises"

Done

Figure 2:

please replace "3 box-car" by "3-month boxcar"

Adopted

Figure 5:

"ozone sonde" or "ozonesonde" : please make it uniform throughout the paper

This figure has been removed.

Figure 10:

Please label the panels with a, b and c. and refer to this in the text.

Done

Figure B2:

Place label the panels with a, b and c. and refer to this in the text.

Done

Lines 407,409,492

Please check Bodecker et al., Boyd et al. and Seidel et al. for doi

Doi Added

Lines 539-545:

WMO 2011,2014,2018 and 2022: citations are not complete

Corrected

---

## Author Response (AR2)

We thank both reviewers for their comments. We have made the technical corrections raised by Reviewer #2. The changes are highlighted in the revised manuscript and are noted as below (in blue and red).

**Reviewer #2**

The authors addressed all comments and made substantial modifications to the regression model and to the trend study as well as substantial revisions to the manuscript. The revised manuscript should be accepted for publication after the following changes have been made:

Line 177 :

Simple linear trend :

In the revised version, it is now clear that the simple linear trend is used to investigate the difference between the homogenized and the uncorrected version of the ozone sonde dataset, and for comparison with Godin-Beckmann et al (2022). MLR is used in the trend comparison with the simulations.

You should make clear why you cannot use the MLR to investigate these homogenized vs uncorrected differences too.

We thank the reviewer's comments. We have clarified that we do not analyse the uncorrected dataset further with the MLR analysis. We believe that the simple linear trend analysis is sufficient to highlight the difference in these two datasets. Instead, we focus on using the MLR analysis to attribute the drivers for the ozone trend in the homogenised ozone data (referred to as the "observed" ozone), which is the main purpose of this study.  Changed are made at lines 177-178, 204-205 in the revised manuscript.

Still a simple linear trend is first compared to a MLR estimated trend on line 189 while the similarity between the MLR and the simple linear trend is made clear only in the following section (Fig 4 and line 205). This should be referred to when the comparison is made.

Thanks for the comments. We would like to point out that in the whole Sect. 3.2 (including line 189 and Fig. 3), we compare the LR trends in homogenised and uncorrected data, without referring to the MLR trend.  The comparison between LR and MRL trends (for homogenised data) are in Sect. 3.3 (after line 203). We hope that we have clarified this with the changes made stated above.

Figures 3 and 4:

The used color scheme is confusing:

Fig 3ab black and Fig 4ab red: simple linear trend on homogenized dataset

Fig 4ab black: MLR trend (with proxys)

The black curve in Fig 3 corresponds to the red curve in Fig 4 and not to the black curve in Fig 4 while the red curve in Fig 3 is FTIR.

This should be adapted.

Thanks very much for the suggestion. We have modified the colours in Fig. 4 to avoid confusion.

line 191:

replace "it exceeds 6-7% decade-1" with "it exceeds -6% decade-1"

Thanks – we have made the change. (line 92)

line 197:

derived instead of drived

Corrected (line 198)

line 221:

"in the stratosphere, the strat…"

instead of

"in the stratosphere. the strat…"

Corrected (line 223)

lines 224 and 350:

positive instead of negative in "by the significant negative linear trend in tropopause height"

(Fig 6c shows a positive trend of tropopause height)

Corrected (lines 226 and 352)

line 357 :

surface instead of Surface in "However, surface"

Corrected (line 359)

line 414 :

"Table 3" instead of "Table ??"

This sentence has been removed from the Appendix B, as Table 3 is now in the main text. (line 419-420)

[revised manuscript text omitted]